# On ozone trend detection: using coupled chemistry-climate simulations to investigate early signs of total column ozone recovery.

James Keeble[1], Hannah Brown[1], N. Luke Abraham[1,2], Neil R. P. Harris[3], and John A. Pyle[1,2]

[1]University of Cambridge, Department of Chemistry, Cambridge, UK
[2]National Centre for Atmospheric Science, Cambridge, UK
[3]Centre for Environmental and Agricultural Informatics, Cranfield University, Cranfield, UK

*Correspondence to*: James Keeble (james.keeble@atm.ch.cam.ac.uk)

**Abstract.** Total column ozone values from an ensemble of UM-UKCA model simulations are examined to investigate different definitions of progress on the road to ozone recovery. The impacts of modelled internal atmospheric variability are
accounted for by applying a multiple linear regression model to modelled total column ozone values, and ozone trend analysis is performed on the resulting ozone residuals. Three definitions of recovery are investigated: (i) a slowed rate of decline and the date of minimum column ozone; (ii) the identification of significant positive trends; and (iii) a return to historic values. A return to past thresholds is the last state to be achieved. Minimum column ozone values, averaged from 60°S-60°N, occur between 1990 and 1995 for each ensemble member, driven in part by the solar minimum conditions
during the 1990s. When natural cycles are accounted for, identification of the year of minimum ozone in the resulting ozone residuals is uncertain, with minimum values for each ensemble member occurring at different times between 1992 and 2000. As a result of this large variability, identification of the date of minimum ozone constitutes a poor measure of ozone recovery. Trends for the 2000-2017 period are positive at most latitudes and are statistically significant in the mid-latitudes in both hemispheres when natural cycles are accounted for. This significance results largely from the large sample size of the
multi-member ensemble. Significant trends cannot be identified by 2017 at the highest latitudes, due to the large interannual variability in the data, nor in the tropics, due to the small trend magnitude, although it is projected that significant trends may be identified in these regions soon thereafter. While significant positive trends in total column ozone could be identified at all latitudes by ~2030, column ozone values which are lower than the 1980 annual mean can occur in the mid-latitudes until ~2050, and in the tropics and high latitudes deep into the second half of the 21st century.

# 1 Introduction

2017 marked the 30[th] anniversary of the Montreal Protocol, which was implemented to protect the stratospheric ozone layer from the harmful effects of ozone depleting substances (ODS). These gases, mostly inert in the troposphere, breakdown when they reached the stratosphere, with the subsequent products then leading to chemical ozone depletion (e.g. Molina and Rowland, 1974; Stolarski and Cicerone, 1974; Rowland and Molina, 1975). Controls introduced under the Montreal Protocol
30   and its subsequent amendments first slowed the rate of accumulation of these halogenated ODS in the atmosphere, and since the late 1990s their atmospheric concentrations have begun to decline (Newman et al., 2006; Mäder et al., 2010; WMO 2011; 2014). A reduction in equivalent stratospheric chlorine (ESC; Eyring et al., 2007) concentrations should lead to an increase in atmospheric ozone as the strength of the halogen catalyzed ozone destruction cycles declines. However, detecting recovery of the stratospheric ozone layer is complicated by a number of additional factors which affect the year to
35   year variability of total column ozone values. These factors include volcanic eruptions, such as the eruption of Mt. Pinatubo in 1991 (e.g. Randel et al., 1995; Telford et al., 2009), changes in the solar cycle (e.g. Brasseur, 1993; Van Loon and Labitzke, 2000; Austin et al., 2007; Calisesi and Matthes, 2007) and variability in ozone resulting from a range of factors

affecting dynamical variability, including the quasi-biennial oscillation (QBO; e.g. Hollandsworth et al., 1995; Baldwin et al., 2001; Leblanc and McDermid, 2001) and variations in sea surface temperatures, particularly those related to the El Niño-Southern Oscillation (ENSO; e.g. Braesicke and Pyle, 2004; Manzini, 2009; Randel et al., 2009). In addition, long term total column ozone trends are driven in part by emissions of other non-chlorinated anthropogenic species, such as $CO_2$, $CH_4$ and

$N_2O$, which affect stratospheric ozone concentrations by altering stratospheric temperatures and dynamics (Haigh and Pyle, 1982; Avallone and Prather, 1996; Plumb, 1996; Eyring et al., 2010; 2013; Iglesias-Suarez et al., 2016), and in the case of $CH_4$ and $N_2O$ by acting as source gases for reactive $HO_x$ and $NO_x$ species (Chipperfield and Feng, 2003; Ravishankara et al., 2009; Revell et al., 2012; Meul et al., 2014). Identification of significant trends is also made problematic by the difference in year-to-year variability in total column ozone values in different regions. For example, high northern latitudes exhibit

very large interannual variability in winter and spring, while variability in the Southern Hemisphere is comparatively smaller. Furthermore, there is a dynamical response to changes in chemical ozone depletion in the stratosphere, which may enhance/impede future recovery by altering the transport of ozone (e.g. McLandress et al., 2011; Braesicke et al., 2013; Keeble et al., 2014). In comparison, the chemical ozone depletion signal in the tropics is small and total column ozone variability is dominated by features such as the solar cycle, QBO and ENSO. As a result of all of these factors, identifying

robust recovery of total column ozone and ascribing that recovery to a decline in stratospheric halogen species is a complex issue.

For past trends, recovery of the stratospheric ozone layer could be detected using two different methodologies: process oriented studies and statistical analysis of datasets. For the first, observations can be compared with a detailed chemistry-transport model which includes all known processes. If good agreement is found between the model and observations when

all processes are included, then evidence of ozone recovery due to decreasing stratospheric halogen loadings can be identified by excluding other processes. For example, Solomon et al. (2016) found evidence for healing of the Antarctic ozone layer in September when polar halogen chemistry is included but interannual dynamical variability and volcanic factors are excluded. For the second method, a statistical approach can be followed in which data are used to detect significant change between time periods. The impact of confounding changes (e.g., QBO, solar cycle, etc.) can be quantified

using multiple linear regression and removed from the statistical analysis of the data in order to provide a better estimate of long-term trends (e.g. Staehelin et al., 2001; Reinsel et al., 2005; WMO, 2007; Harris et al., 2015; Chipperfield et al., 2017). These statistical approaches nearly all work by relying on the assumption of a linear relationship between a proxy variable and its impact on total ozone. Using this method, a number of recent studies has started to explore if observed total column ozone and ozone profile values show signs of recovery (e.g. Pawson et al., 2014; Harris et al., 2015; Steinbrecht et al., 2017;

Ball et al., 2018; Weber et al., 2018). These studies have indicated that statistically significant recovery of column ozone values can be identified in some datasets at some latitudes, but that this is not true for all datasets (e.g. Weber et al., 2018). As recovery trends are calculated over relatively short time frames (<20 years), identification of trend magnitude and trend significance from observations can be affected by high or low values at the beginning or end of the observational record (compare, for example, the trends derived by Pawson et al. (2014) with those of Weber et al. (2018)).

To explore future ozone trends and recovery, data from coupled chemistry-climate model (CCM) simulations are required. Each CCM simulation constitutes a possible future evolution of stratospheric ozone. In order to sample the effect of internal atmospheric variability on ozone and to derive an estimate of uncertainty of future trends, multiple ensemble members can be run in which the initial conditions of each simulation are modified but the same forcings are prescribed (e.g. GHG (greenhouse gas) evolution, aerosol loadings). Greater confidence can be assigned to significance of the mean trend as the

number of ensemble members increases. Multiple ensemble members also give information about the possible range of future trends and as a result are not as sensitive to high or low values at the beginning or end of the record of any individual

ensemble member, in contrast to single member simulations and observational records. Thus, using an ensemble of future projections from a single CCM can provide additional insight into the detection of different phases of ozone recovery.

In this study, we use results from a chemistry-climate model coupled with statistical approaches to explore different definitions of ozone recovery (see Reinsel et al., 2005; Weatherhead and Andersen, 2006; Chipperfield et al., 2017). In
particular we define three stages of total column ozone recovery:

1. A reduced rate of decline in ozone and the date of minimum ozone.
2. Statistically significant increases in column ozone values, after accounting for natural variability, that can be ascribed to reductions in ESC.
3. Return of total column ozone values to some specified past value (typically 1980 or 1960).

Identifying when and if each of these stages has occurred at different latitudes, and being able to assess the confidence with which this can be done, is fundamental to determining the success of the Montreal Protocol. For this work we use the ozone fields calculated in an ensemble of UM-UKCA transient simulations, which are described in section 2. We carry out a statistical analysis of the model results, as outlined in section 3, to identify when each of these stages of recovery occurs for different latitude ranges. These results are presented in section 4, 5 and 6 and implications are discussed in section 7.

**2 Model configuration and simulations**

An ensemble of transient simulations was performed using version 7.3 of the HadGEM3-A configuration of the Met Office's Unified Model (Hewitt et al., 2011) coupled with the United Kingdom Chemistry and Aerosol scheme (hereafter referred to as UM-UKCA). This configuration of the model has a horizontal resolution of 2.5° latitude by 3.75° longitude, with 60 vertical levels following a hybrid sigma-geometric height coordinate with a model top at 84 km. The chemical scheme used
in this configuration of the model is an expansion of the scheme presented in Morgenstern et al. (2009) in which halogen source gases are considered explicitly, resulting in an additional 9 species, 17 bimolecular and 9 photolytic reactions. Stratospheric aerosol concentrations are prescribed using a climatology based on observations (from SPARC, 2006; described by Eyring et al., 2008) for the historical part of the run, after which background concentrations of stratospheric aerosol loadings are prescribed. HadGEM3-A includes an internally generated quasi-biennial oscillation (QBO), which in
this configuration of the model has a period of ~27 months while the magnitude of modelled easterly(westerly) equatorial zonal wind speed is ~25 m s$^{-1}$ (10m s$^{-1}$), both aspects in good agreement with observed zonal winds at Singapore (e.g. Lee and Smith, 2003). The configuration of the model used for this study includes the effects of the 11-year solar cycle in both the radiation and photolysis schemes. The top of atmosphere solar flux follows historical observations from 1960 to 2009, after which a repeating solar cycle is imposed which is an amplitude equivalent to the observed cycle 23 (as detailed in
Bednarz et al., 2016).

The transient simulations were performed following the experimental design of the WCRP/SPARC CCMI REF-C2 experiment (Eyring et al., 2013), which adopts the RCP6.0 scenario for future GHG and ODS emissions. Two of these ensemble members were run from 1960 to 2099 and an additional five were run from 1980 to 2080. All ensemble members have identical time-dependent boundary conditions, but differ in their atmospheric initial conditions, thereby providing an
estimate of internal atmospheric variability. The simulations were performed in an atmosphere-only configuration, and each ensemble member uses prescribed sea surface temperatures and sea ice fields taken from a parent coupled atmosphere-ocean HadGEM2-ES simulation as lower boundary conditions. The simulations used for this study are described in more detail in

Bednarz et al. (2016) and Keeble et al. (2017), and were performed in support of phase 1 of the Chemistry–Climate Model Initiative (CCMI; Morgenstern et al., 2017).

## 3 Removing natural cycles

Identifying an increase in total column ozone resulting from reductions in stratospheric chlorine requires removing the
effects of natural processes, such as volcanic eruptions, the QBO, ENSO and solar cycle, from the modelled total column ozone data, as these cycles may impose short terms trends in the data which are wrongly interpreted as signs of recovery.  In order to identify the impacts of these natural processes on modelled total column ozone we create a statistical model using multiple linear regression (MLR) analysis.  This process assumes that the modelled total column ozone values, TO3, can be reproduced by combining some constant value of total column ozone, *i*, which corresponds to the intercept term of the MLR,
with a number of explanatory, or predictor, variables.  This statistical model can be expressed as:

$$TO3_{e,l,t} = i_{e,l} + \alpha_{e,l}^{QBO_{50}}.QBO_{50_{e,t}} + \alpha_{e,l}^{QBO_{30}}.QBO_{30_{e,t}} + \alpha_{e,l}^{solar}.solar_t + \alpha_{e,l}^{ENSO}.ENSO_t + \alpha_{e,l}^{aerosol}.aerosol_t$$
$$+ \alpha_{e,l}^{ESC}.ESC_{e,l,t} + N_{e,l,t}$$

in which the α values are the coefficients returned from the MLR for each explanatory variable (denoted by the superscript) and vary between latitude range and ensemble member.  The explanatory variables included in the MLR are the QBO, solar cycle, ENSO, volcanic aerosols and ESC.  The subscripts *e*, *l* and *t* indicate that the alpha value or explanatory variable differs with ensemble member, latitude and/or time respectively.  For the QBO, two terms are included, $QBO_{50}$ and $QBO_{30}$,
which correspond to equatorial westerly winds at 50 hPa and 30 hPa respectively.  Two QBO terms are included to account for the phase shift in the total column ozone response with respect to QBO changes at different altitudes.  The solar cycle is represented by the top of atmosphere solar flux, represented in the MLR as *solar_t*.  ENSO effects on column ozone are represented by *ENSO_t*, the detrended sea surface temperature anomalies in the NINO3.4 region.  Volcanic aerosols are included as hemispheric aerosol optical depths, and so are different for the northern and Southern Hemispheres to account
for the lack of interhemispheric transport of aerosols emitted into the stratosphere from high latitude eruptions.  The final term included in the MLR, *ESC*, represents stratospheric chlorine concentrations.   This term is equal to the ESC concentration at 30 km for each latitude bin to account for the time taken for ODS to be transported to higher latitudes.  Any month to month variation not accounted for by the explanatory variables in the MLR is represented by the noise term $N_{e,l,t}$.  The *Solar_t*, *ENSO_t* and *aerosol_t* terms are all prescribed forcings in the model and do not vary between ensemble members.
In this study we use deseasonalised monthly mean total column data, and so there is no need for a seasonal cycle term in the MLR model.

This statistical model can then be used to remove the component of total column ozone variations related to the QBO, solar cycle and volcanic aerosol changes from the raw model data to leave a set of ozone residuals, RO3, which retain the long-term trend and any interannual variability not explained by the MLR:

$$RO3_{e,l,t} = TO3_{e,l,t} - \left( \alpha_{e,l}^{QBO_{50}}.QBO_{50_{e,t}} + \alpha_{e,l}^{QBO_{30}}.QBO_{30_{e,t}} + \alpha_{e,l}^{solar}.solar_t + \alpha_{e,l}^{ENSO}.ENSO_t + \alpha_{e,l}^{aerosol}.aerosol_t \right)$$

MLR analyses and ozone residuals are produced for each individual ensemble member of the raw model data, resulting in seven RO3 time series.  For both the raw model data and the ozone residuals, decline and recovery trends are calculated using independent linear trend fits for the periods 1980-1997 and 2000-2017 respectively.  When calculating trends for both the raw model data and ozone residuals, a single linear fit is produced using data from all seven ensemble members rather than producing a fit for each individual ensemble.  In order to make any robust conclusions about the statistical significance

of modelled trends, some measure of the trend uncertainty is required. Here we calculate trend uncertainties following the methodology of Weatherhead et al. (1998), in which the standard deviation of the uncertainty in the linear trend is calculated by:

$$\sigma_{trend} = \frac{\sigma_{data}}{n^{3/2}} \sqrt{\frac{1 + \varphi}{1 - \varphi}}$$

where $\sigma_{data}$ is the standard deviation of the time series in question (either raw model data or RO3), $n$ is the number of months in the time series and $\varphi$ is the autocorrelation coefficient for a 1 month lag. As discussed by Weatherhead et al. (1998) autocorrelation can be substantial for monthly mean time series, particularly in low latitudes, and failure to account for it results in an underrepresentation of trend uncertainty. As for the trend calculations, trend uncertainties are calculated using data from all seven ensemble members together, and as such for the 17 year periods considered for the decline and recovery phases $n$ is very large ($n$=17x12x7=1428).

## 4 Modelled global column ozone and minimum values

Figure 1 shows deseasonalised monthly mean total column ozone anomalies relative to 1980 values, averaged over 60°S-60°N, from 1960 to 2100 for each individual ensemble member (light blue lines) and the ensemble mean (dark blue line). A sharp decrease in total column ozone is modelled from 1980 to the late 1990s, consistent with increased ESC loadings resulting from the use and emission of ODS. From the late 1990s until ~2070 column ozone values gradually increase, exceeding their 1980s values by ~2030, and their 1960s values by ~2050. Beyond 2070 total column ozone values remain relatively constant until the end of the century. Superimposed on these long term trends is the effect of the solar cycle, which imprints a distinctive 11-year oscillation in the data. Alongside the modelled total column ozone anomalies are shown values from version 2.8 of the Bodeker Scientific total-column ozone dataset in black (Bodeker et al., 2005). There is generally good agreement between modelled total column ozone anomaly values and the Bodeker dataset; decadal total column ozone changes, the response of column ozone to the solar cycle and the magnitude of interannual variability are all well captured by the model ensemble throughout the time period during which the observations and model data overlap.

Also shown in Figure 1 are the ozone residuals calculated when the effects of natural cycles are removed, as detailed in section 3 (red lines). This dataset follows the long term trends of the raw UM-UKCA data, but the cyclic short term trends in column ozone values have been removed. Most obvious from Figure 1 is the removal of the 11-year solar cycle signal, leading to a much smoother, monotonically increasing trend from 2000 to 2060 compared to the raw model data.

As discussed above, the first signs of detectable ozone recovery would be identified as a reduced rate of decline in column ozone and the date of minimum ozone. Modelled total column ozone values generally decrease from 1980 to the late 1990s (blue line, Figure 1), consistent with the increase in ESC amounts. However, this decrease is not constant; rapid decline is modelled from 1980 to 1985 and from 1990 to 1995, while between these periods total column ozone abundances are relatively constant, or even increase (see inset in Figure 1). This feature is also seen in the Bodeker dataset, and predominantly results from the impact of the solar cycle on stratospheric ozone concentrations. As top of atmosphere solar flux decreases from solar maximum to solar minimum, rapid decline of total column ozone occurs as this effect combines with the impacts of increasing ESC. Conversely, as top of atmosphere solar flux increases, enhanced stratospheric ozone production temporarily offsets the chemical ozone destruction resulting from increased ESC concentrations. This is confirmed by analyzing the ozone residuals (red line Figure 1), which show a much smoother decline from 1980 to the late

1990s, and highlights the importance of understanding the drivers of short term trends in raw total column ozone values when trying to assess longer term trends.

As well as influencing the trajectory of declining column ozone abundances, natural cycles also affect the timing and magnitude of minimum total column ozone values. In the raw model data, the minimum total column ozone values averaged over 60°S-60°N are reached between 1992-1994, depending on the ensemble member, which is several years before the peak loading of ESC in 1997 (e.g. Mäder et al., 2010; WMO 2011; 2014). This offset in timing between peak ESC and total column ozone minimum results from the impact of the solar cycle, as discussed above, and the eruption of Mt. Pinatubo on total column ozone. The early 1990s was a time of low top of atmosphere solar flux, while the eruption of Mt. Pinatubo increased stratospheric sulphate surface area density, both reducing total column ozone abundances. When the effects of these natural cycles are removed, ozone residuals (red line Figure 1) are larger than modelled total column ozone values throughout the early 1990s.

Although this work indicates minimum column ozone values occurred in the 1990s, this is a poor metric for making robust conclusions about ozone recovery. Firstly, the ozone minimum may occur because there is no more capacity for increased chemical depletion despite increasing ESC. This is the case over Antarctica during springtime during the 1990s, where near complete destruction of polar lower stratospheric ozone occurs and any additional increase in ESC would have a negligible effect. Secondly, minimum column ozone values are very sensitive to dynamical conditions. For example, Bednarz et al. (2016) have shown that even under much lower stratospheric halogen loadings significant ozone depletion can occur in the Arctic lower stratosphere during conditions which favour a cold, stable polar vortex. Even outside the high latitudes, where interannual variability in total column ozone values is largest, identification of the year of minimum ozone is uncertain, with each of the seven residual ozone time series having minimum values at different times between 1992-2000 (light red lines, Figure 1).

## 5 Regional trends

Decline and subsequent recovery of total column ozone is often calculated using linear trends for two periods either side of an inflection time (e.g. Newchurch et al., 2003; Reinsel et al., 2005; Jones et al., 2009; Nair et al., 2013; Chehade et al., 2014). Previous studies have identified 1997 as the inflection time for long-term total column ozone observed trends (e.g. Harris et al., 2008), and as a result we define the decline phase as 1980-1997 with the recovery phase defined from 2000-2017. Here we calculate independent linear trends for both the decline and recovery phases firstly using the raw total column ozone data from the UM-UKCA model (discussed below) and then using model data in which the effects of the natural processes discussed above have been removed using the statistical model introduced in Section 3.

Figure 2 shows total column ozone trends (in DU year$^{-1}$) obtained from both the raw data from the UM-UKCA simulation and the ozone residuals for the decline (1980-1997) and recovery (2000-2017) phases, averaged over 10° latitude bands. Error bars associated with each trend represent the 95% confidence intervals ($2\sigma_{trend}$), calculated as described in section 3. During the decline phase, ozone trends for both the raw model data and ozone residuals are greatest at high latitudes due to the heterogeneous activation of chlorine on PSCs within the polar vortex and the transport of mid-latitude ozone depletion signals to high latitudes by the BDC. The uncertainty associated with the trends is also largest at high latitudes, due to the higher year-to-year variability in chemical and dynamical processes at high latitudes compared with the tropics. Negative trends in the raw column ozone data from 1980-1997 are significant at all latitudes, although when natural cycles are removed the trends from 10°S-10°N are not significant. At all latitudes there is a more negative trend in the raw UM-UKCA data compared with the dataset in which the natural cycles have been removed. This is the result of the eruption of Mt

Pinatubo and the pronounced solar minimum during the 1990s, both of which resulted in lower column ozone values and so a greater trend from 1980. This can be clearly seen in Figure 1 by comparing the blue and red lines. However, the close agreement in trends between the raw modelled values and the ozone residuals calculated when natural cycles are accounted for indicates that natural cycles have had only a small contribution to the trends during the period 1980-1997 (consistent with the findings of Gillett et al., 2011). Trends for the decline phase, calculated for both the raw model data and ozone residuals, agree within the uncertainty estimates with those obtained from observation datasets by Weber et al. (2018) and those obtained from CCMVAL2 models (e.g. Pawson et al., 2014).

When considering the recovery phase, positive trends are modelled at all latitudes from 2000-2017 for both the raw model data and ozone residuals. For the raw model values, these trends are only significant at the 95% confidence interval in the Southern Hemisphere between 80°S-50°S. However, when the effects of natural cycles are removed from the data, significant positive trends can be identified in the Southern Hemisphere between 80°S-30°S, and also in the Northern Hemisphere from 20°N-70°N. Significant trends cannot be identified by 2017 at the highest latitudes, in either dataset, due to the large interannual variability in the data, nor in the tropics due to the small trend magnitude. As for the decline phase, while accounting for nature cycles in the ozone residuals reduces the trend uncertainties, it does not significantly affect the trend magnitudes, indicating that natural cycles do not significantly contribute to recent increases in column ozone values.

Trends for the recovery phase, calculated for both the raw model data and ozone residuals, are consistent with those calculated for observational datasets by Chehade et al. (2014), Pawson et al. (2014) and Weber et al. (2018), and agree with all three studies within trend uncertainty estimates. In general, while in agreement with each of these studies, trends presented here are larger than those presented by Weber et al. (2018), and are closer in magnitude to the findings of Chehade et al. (2014) and Pawson et al. (2014). The modelled trends presented here indicate, as Weber et al. (2018) conclude, that the differences between the latest trends calculated from observations by Weber et al. (2018) and the earlier trends estimated by Pawson et al. (2014) result from lower than average column ozone values at the end of the observational record, which are part of natural interannual variability and likely do not represent some fundamental shift in the trajectory of ozone recovery.

Identification of significant trends depends on the gradient of the trend, the number of data points (in this case the number of modelled monthly means) and the variance and autocorrelation of the data (e.g. Weatherhead et al. 2000). Analyzing the near global (60°S-60°N) raw total column ozone data (blue lines, figure 1), year 2000 is a solar maximum year and so total column ozone values are relatively high compared to the following few years. It is not until ~11 years later, during the next solar maximum, that trends become positive. Trend analysis on data between 2000 and 2015 could indicate that there is a significant positive trend, which could in turn lead to the conclusion that significant recovery of the ozone layer had begun. However, as further years are considered, from 2015 to 2020, total column ozone values start to decline as the solar cycle moves towards a solar minimum, and the magnitude of the recovery trend is reduced while the variance in the residuals increases. Now, trends calculated from 2000 to 2020 are no longer statistically significant. As a result, when assessing recovery trends it is necessary to use datasets in which the effects of natural cycles have been accounted for, such as the ozone residuals calculated in section 3. However, as the MLR analysis performed on the raw data does not capture the full modelled interannual variability (hence the occurrence of the noise term, $N_{e,l,t}$), trend magnitudes calculated using ozone residuals from any individual ensemble member are similarly affected by anomalously low or high values at the start or end of the time series. The use of multiple member ensemble simulations in this study mitigates this effect.

The impacts of the noise term on trend magnitudes for each of the individual ensemble members is explored in more detail in figure 3, which shows the year after 2000 at which trend significance can be identified in the ozone residuals for either the

first time (blue), or final time (red). The year a trend becomes significant can be calculated for each ensemble member by identifying the first month after January 2000 in which trends become significant (initial recovery) and the month after which they remain significant (robust recovery). Here trend significance is calculated using the trend uncertainty obtained when all ensemble members are used, as discussed above, but the trend magnitude is calculated for each ensemble member

individually so as to provide a range of dates that trend significance can be identified. This range of dates reflects the impact of unaccounted for noise on the trend magnitude, while maintaining significance thresholds which are consistent between figures 2 and 3. Error bars on figure 3 represent the 95% confidence intervals, calculated as twice the standard deviation of the seven values obtained for the year trend significance is identified (one for each ensemble member).

Distinguishing between initial and robust recovery significance dates is necessary since, as discussed above, trends can be

significant after a number of months and then become non-significant as more data is added so that the variance in the data can increase or the magnitude of the trend can decrease. The first instance of detection of significant trends can be considered as false recovery if it does not coincide with the time after which trends never become non-significant. Note that if the MLR described in section 3 accurately represented all drivers of interannual variability (i.e. the $N$ term was zero), there would be no distinction between initial and robust recovery.

For the ensemble of ozone residuals presented here, mid-latitude trends become significant earlier than those of the tropics or high latitudes. This is due to the high degree of interannual variability at high latitudes, particularly in the Arctic, and the small magnitude of the trends in the tropics. Therefore, it is likely that both initial and robust recovery will first be observed in the mid-latitudes. In addition, both measures of recovery occur at similar times, minimizing the risk of identifying false recovery. Correct identification of robust recovery is important when considering observations of total column ozone and

highlights the need to treat detection of significant recovery for the first time with caution as additional months/years of observational data may reduce the statistical significance of any observed trends. It should be noted that no individual ensemble member shows statistically significant recovery by 2017, and only when considering data from all the ensembles together when calculating trend uncertainties can significant trends be identified.

Once the difference in projected recovery trends is accounted for, these findings are consistent with those of Weatherhead et

al. (2000), who also identified the mid-latitudes as the best location to identify early signs of ozone recovery. However, there is an offset in when trends are expected to become significant between the two studies, with projected detection years modelled in this study generally occurring earlier than those of Weatherhead et al. (2000), particularly in the tropics. This discrepancy in the tropics is unsurprising, as recovery of the tropical ozone column is dependent on the competing influences of declining CFCs, decreasing stratospheric temperatures and changing BDC speeds (see e.g. Eyring et al., 2013; Meul et al.,

2016; Keeble et al., 2017), with increases to the BDC offsetting ozone recovery in the lower stratosphere and resulting in smaller column ozone recovery trends. There is poor agreement in modelled projections of future BDC speeds, and as a result projections of tropical column ozone differ significantly between models (e.g. WMO, 2011). In addition, ozone depletion in the tropics resulting from increasing ESC concentrations is weak in comparison to the mid and high latitudes, and as a result identifying significant recovery trends is particularly sensitive to any interannual variability not accounted for

in the MLR, which may differ between models.

## 6 Return to historic values

While identification of statistically significant increases in total column ozone is a real sign that ozone recovery is occurring, recovery can be said to be complete when column ozone values reach their pre-CFC values again. Traditionally these return thresholds are taken to be either 1980 or 1960 values; here we use 1980. It is likely that future total column ozone values

will initially exceed the 1980s threshold and then fall below this value again due to interannual variability and the effects of the solar cycle and QBO. As a result two metrics are considered: the first time total column ozone exceeds the 1980s threshold, and the last time total column abundances are below the threshold. Between the two time periods total column ozone values rise above and fall below the threshold.

Figure 4 shows the year in which raw total column ozone abundances return to their 1980s values for the first time (blue) and final time (red) for each 10 degree latitude bin. In the tropics, total column ozone exceeds the 1980s threshold as early as 2000 as the amplitude in total column ozone variations resulting from the solar cycle is greater than the decrease in total column ozone resulting from ESC changes. However, despite this region seeing the first values greater than those of the 1980s, it is the only region in which total column ozone abundances are not greater than their 1980s values by the end of the

simulation, consistent with other studies (e.g. Eyring et al., 2013; Meul et al., 2016). This is due to decreasing lower stratospheric ozone concentrations resulting from an acceleration of the BDC under increased greenhouse gas concentrations offsetting increased upper stratospheric ozone concentrations due to decreased ESC and increased $CO_2$ (explored in detail in Keeble et al., 2017).

In the Northern Hemisphere mid-latitudes earliest recovery occurs by ~2020, while final recovery occurs by 2040. The

closeness of these two dates is due to the large trend to variability ratio in the mid-latitudes compared to both the tropics and Arctic. The results are similar in the Southern Hemisphere mid-latitudes, although both dates are delayed by around 10 years, most likely due to the effects of Antarctic polar ozone depletion and transport of ozone poor air masses into these latitudes upon the collapse of the Antarctic polar vortex.

Earliest recovery to historic values at high southern latitudes occurs by ~2040, with final recovery occurring by 2060.

However, the signature of this recovery is very sensitive to calendar month, and earlier signs of recovery may be identified in certain months (e.g. September; Solomon et al., 2016). Arctic column ozone exhibits high interannual variability, with values exceeding the 1980s threshold as early as 2010. However, final recovery is not projected until ~2060.

The future evolution of total column ozone is dependent on the emissions scenario considered, and the exact timings of recovery to historic values will vary with changes to $CO_2$, $N_2O$ and $CH_4$ emissions as well as ESC reductions. As a result,

the expected return dates for each latitude will evolve as we approach those dates, in line with our increased understanding of the emissions pathway or if future emission controls come into effect.

**7 Discussion and Conclusions**

In this study we analyse total ozone values from an ensemble of UM-UKCA model runs to investigate different definitions of progress on the road to ozone recovery. . In particular, we have investigated three definitions: (i) a slowed rate of decline

and the date of minimum ozone; (ii) the identification of significant positive trends; and (iii) a return to historic values. The impacts of natural cycles on modelled internal atmospheric variability are accounted for by applying a multiple linear regression model to modelled total column ozone values. The use of multi-member CCM ensembles, in which each simulation constitutes a possible future evolution of stratospheric ozone, allows us to better account for the modelled internal atmospheric variability not captured by the explanatory variables in the MLR, and so provide greater confidence when

assessing the statistical significance of each definition.

The first and most obvious conclusion is that recovery can be identified in the first two metrics before a return to past thresholds is achieved. For the first definition of recovery, minimum total column ozone values averaged from 60°S-60°N occur between 1990 and 1995 for each ensemble member, driven in part by the solar minimum conditions during the 1990s.

When natural cycles are accounted for, identification of the year of minimum ozone in the resulting ozone residuals is uncertain, with minimum values for each of the seven residual ozone time series occurring at different times between 1992 and 2000. As a result, identification of the date of minimum ozone values is problematic and a poor measure of ozone recovery.

For the second definition of recovery, positive trends are modelled at all latitudes from 2000-2017 for both the raw model data and ozone residuals. In contrast to recent analysis of total ozone measurements (e.g. Chipperfield et al., 2017; Weber et al., 2018), when the effects of natural cycles are removed from the data, statistically significant positive trends can be identified in the Southern Hemisphere between 80°S-30°S, and also in the Northern Hemisphere from 20°N-70°N. This increased significance results largely from the much larger sample size that arises in a multi-member ensemble and the

resulting reduction in the uncertainty associated with the mean trend. Significant trends cannot be identified by 2017 at the highest latitudes, due to the large interannual variability in the data, nor in the tropics, due to the small trend magnitude. It was found that while accounting for the effects of natural cycles impacted trend uncertainty estimates, it did not significantly affect trend magnitudes for either the decline or recovery phases, indicating that natural cycles have played only a minor role in recent trends, consistent with previous studies.

It is important to note that, while a statistically significant, positive recovery trend could be calculated at a particular point of time, additional years of data may lead to a reduced significance of trend, due to either a decrease in the magnitude of the trend or an increase in interannual variability. This effect results in a need to distinguish between initial recovery (the time at which trends become significant for the first time) and robust recovery (the time after which trends remain significant despite adding further years). Accounting for this, we identify the mid-latitudes as the best place to find early signs of ozone

column recovery. This is due to the combination of reasonably large trend magnitudes and comparably low variability (especially in the Southern Hemisphere). Despite the large trend magnitudes modelled in the high latitudes, interannual variability in these regions resulting from both the large dynamic interannual variability and the large changes in chemical ozone loss occurring from year to year is too large for a statistically significant signal to be easily detected. In contrast, the small trend magnitudes modelled in the tropics confound identification of statistically significant ozone recovery.

For the third definition of recovery, a return to historic values, it was found that, while robust recovery could be identified at all latitudes by ~2030, column ozone values which are lower than the 1980 annual mean can occur in the mid-latitudes until ~2050, and in the tropics and high latitudes deep into the second half of the 21$^{st}$ century. While projected column ozone values for the mid and high latitudes reach a point after which they are never lower than the 1980 annual mean, consistent with the projected super recovery of ozone, column ozone values lower than the 1980 annual mean occur in the tropics until

the end of the period analysed in this study. This results in part from the large amplitude ozone response to natural cycles, particularly the solar cycle, and also the effects of increased BDC speeds offsetting column ozone recovery resulting from decreased CFCs, as discussed by Keeble et al. (2017).

This work further highlights the need to ensure that the impacts of natural cycles (e.g. solar cycle, QBO, ENSO) on total ozone are correctly described when performing MLR analysis. This is a challenge because of a number of factors. Firstly,

the assumption in all MLR analysis of a linear relationship between the proxy variables used and the impact on ozone is not accurate, and there is growing evidence that these cycles are not isolated, but interact with one another (e.g. White and Liu, 2008; Calvo et al., 2009; Gray et al., 2010). Secondly, cycles with varying amplitudes (e.g. the solar cycle, which shows differing top of atmosphere solar flux during the last four solar maximums) or lengths (e.g. the QBO, the period of which may change in the future and has recently been observed to undergo rapid, non-periodic reversal) have different impacts on

total column ozone which makes accurate estimates of the coefficients for these variables in the MLR harder to achieve.

Finally, volcanic eruptions are particularly difficult to account for in the MLR, both because of the infrequent, non-periodic timings of eruptions, and because eruptions have very different impacts on stratospheric ozone when stratospheric ESC concentrations are high compared to when ESC is low (e.g. Tie and Brasseur, 1995).

Our analysis has focused solely on interpreting the total ozone column record. Many studies have recently examined the trends in the vertical distribution of ozone since ESC maximized (e.g. Harris et al., 2015; Steinbrecht et al., 2017; Ball et al., 2018). In these studies, factors such as higher variability, greater uncertainties and poorer data quality add to the uncertainty in detection of significant trends compared to the total column. However similar studies to this one using ensembles of model runs could provide real insights into the issue, especially in the climatically important lower stratosphere where ozone may still be decreasing (e.g. Ball et al., 2018). As a result we recommend the use of multi-member ensemble simulations, in conjunction with ongoing observational efforts, to better identify signs of ozone recovery for both the total column and ozone profiles.

**Data availability**

Data from the two 1960-2100 transient simulations are available as part of the CCMI initiative through BADC: https://blogs.reading.ac.uk/ccmi/badc-data-access/. All further data are available upon request.

**Competing interesting**

The authors declare that they have no conflict of interest.

**Acknowledgements**

The research leading to these results has received funding from the European Community's Seventh Framework Programme (FP7/2007 - 2013) under grant agreement n° 603557 (StratoClim) and the European Research Council through the ACCI project (project number: 267760). We thank NCAS-CMS for modelling support. Model simulations have been performed using the ARCHER UK National Supercomputing Service and MONSooN system, a collaborative facility supplied under the Joint Weather and Climate Research Programme, which is a strategic partnership between the UK Met Office and the Natural Environment Research Council. We would like to thank Greg Bodeker of Bodeker Scientific, funded by the New Zealand Deep South National Science Challenge, for providing the combined total column ozone database.

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

**Figure 1:** Deseasonalised total column ozone anomalies (in DU) relative to the 1980 mean, averaged over 60°S-60°N, for the seven UM-UKCA transient ensemble members (light blue lines) and ensemble mean (dark blue line). Also shown are
5 the ozone residuals calculated when natural cycles are removed from each ensemble member (light red lines) and the mean of the ozone residuals (dark red line). The inset shows total column ozone anomalies for the transient UM-UKCA simulations and v2.8 of the Bodeker dataset (Bodeker et al., 2005; black line) from 1975 to 2015.

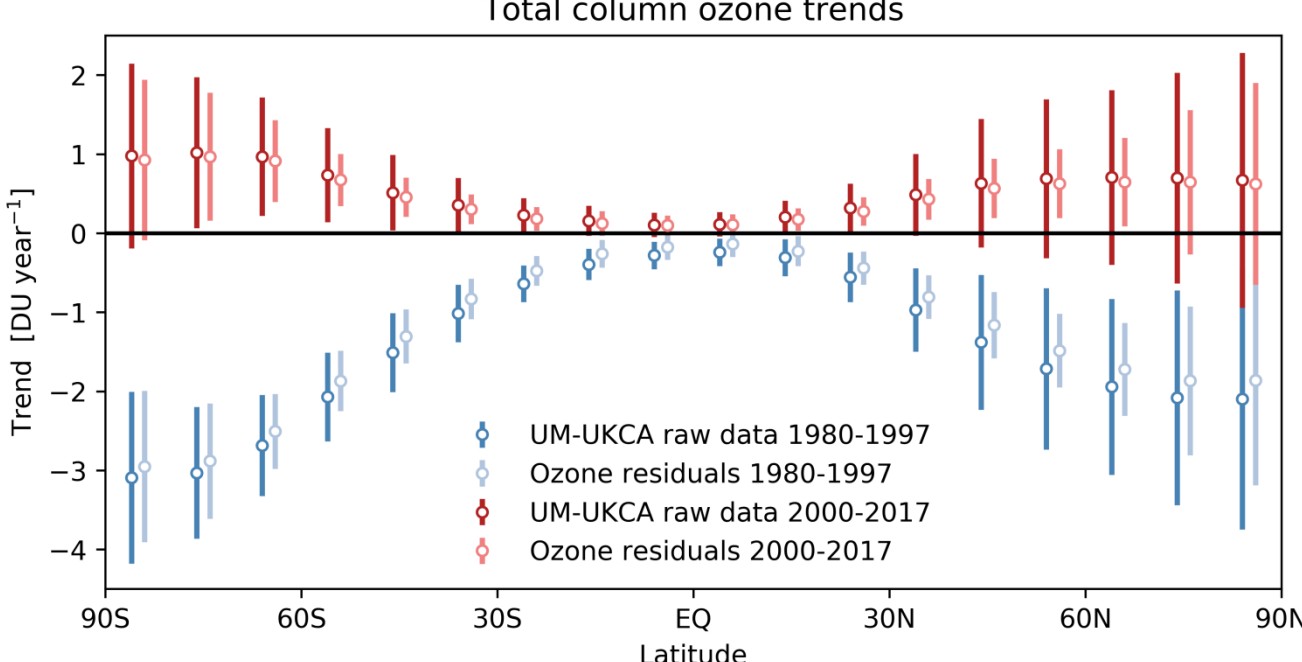

**Figure 2:** Ozone trends from 1980-1997 (blue points) and 2000-2017 (red points) for the raw UM-UKCA data (dark points) and ozone residuals calculated when the effects of natural cycles are removed (light points). Error bars associated with each trend represent the 95% confidence intervals, calculated as described in section 3.

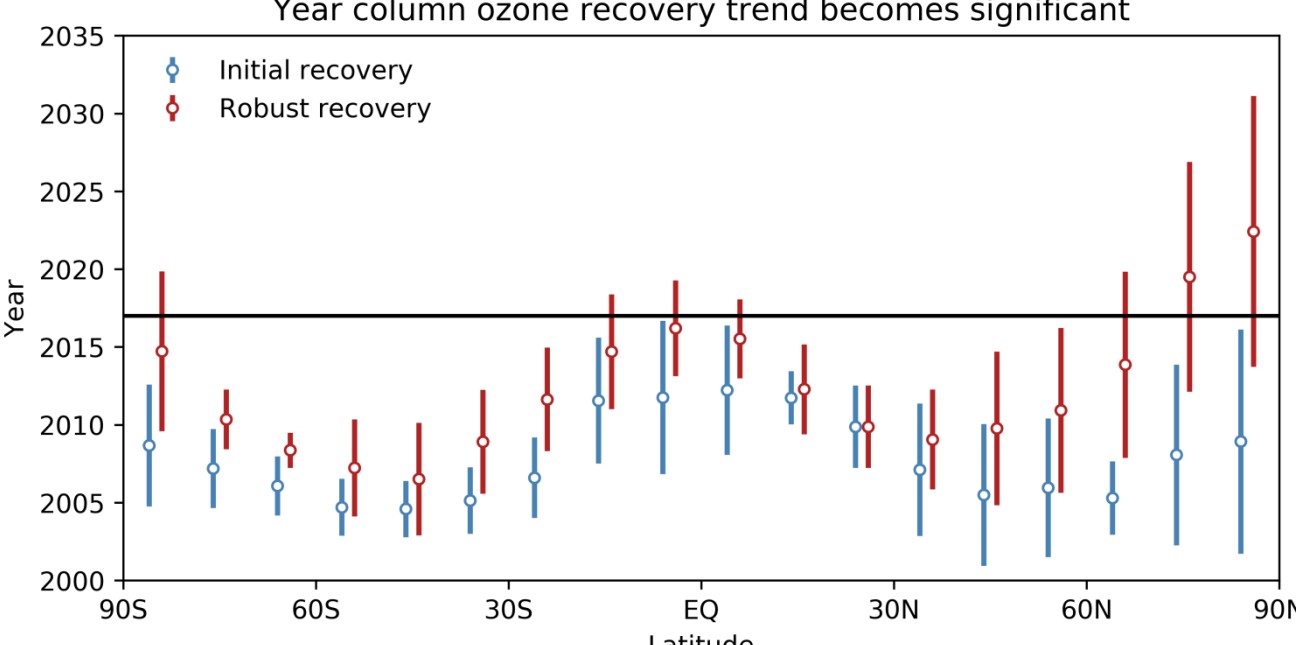

**Figure 3:** Year at which recovery trend of the ozone residuals becomes significant. A distinction is made for the first time significance can be determined (blue points) and the time after which trends remain significant (red points). Here trend significance is calculated using the trend uncertainty obtained when all ensemble members are used, as discussed in section 3, but the trend magnitude is calculated for each ensemble member individually to reflect the impact of unaccounted for noise on the trend magnitude. Error bars represent the 95% confidence intervals, calculated as twice the standard deviation of the seven values obtained for the year trend significance is identified (one for each ensemble member)

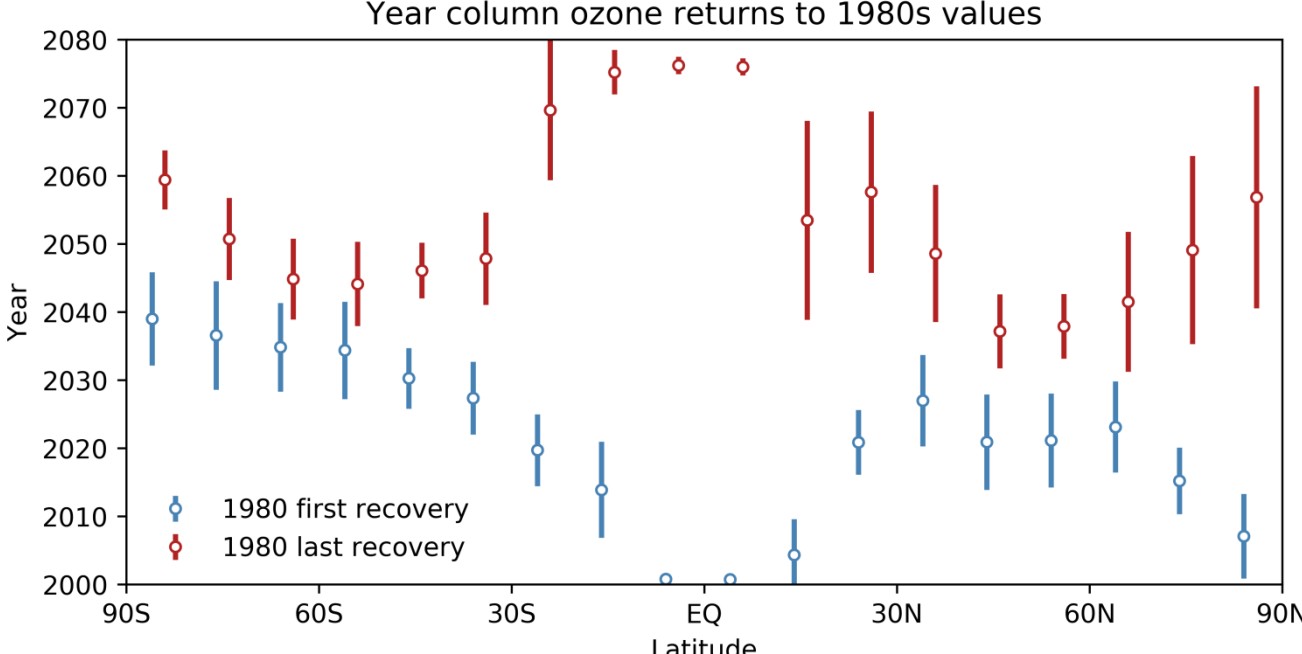

**Figure 4:** Year modelled total column ozone returns to 1980 annual mean values in each latitude band for the raw data from the seven UM-UKCA ensemble members. Blue points represent the first time annual mean values exceed the 1980 mean, while red points represent the final time annual mean values are lower than the 1980 mean. Error bars represent the 95% confidence intervals, calculated as twice the standard deviation of the return dates calculated for each ensemble member.