# Peer review of "On ozone trend detection: using coupled chemistry-climate simulations to investigate early signs of total column ozone recovery"

_Atmospheric Chemistry and Physics, 2017_

## Referee Comment (RC1) · Anonymous Referee #1 · 5 Jan 2018

**1 Recommendation**

Based on simulations by the UM-UKCA chemistry climate model, the authors investigate past and current trends in total column ozone, the years when the expected increases in total ozone might become significant, and the years when total ozone might return to 1980s levels. These questions are relevant for the expected recovery of the ozone layer, and for checking the success of the international Montreal Protocol protecting the ozone layer.

The used data and methods appear solid. The paper is generally clear, concise and

well written. What I am missing, however, is a more thorough comparison with observed trends, and with existing literature. I am also missing a more detailed explanation, how uncertainties were estimated. I think all this can be fixed in a revised version. After addressing my comments below, the paper should be acceptable for publication in ACP.

**2 Major comments**

1.) I think it is absolutely necessary to compare the simulated trends and their uncertainties with trends from observations. In my Fig. 1 below, I have overlaid observed trends from Fig. 7 of Weber et al. (2017) onto the simulated trends from Fig. 2 of the current manuscript. Note that the trends from Weber et al. are in % per decade, so at higher latitudes they tend to appear smaller compared to the DU per year trends of the current manuscript. Nevertheless, the observed trends appear to be much smaller at mid-latitudes and in both hemispheres. The uncertainties, on the other hand seem to be quite comparable. A comparison like that would be extremely valuable. I urge the authors to add such a comparison to their paper. Preferably this would be in an additional Figure, comparing simulated and observed trends (e.g. from Weber et al. 2017), as well as their uncertainties, and using the same units (DU per time or % per time). If this comparison confirms the impression from my Fig. 1 below, and the observed trends at mid-latitudes are indeed much smaller, this would be an important finding. Such an apparent lack of significant ozone increases at mid-latitudes would question our expectations for ozone recovery (see also Ball et al. 2017).

2.) A similar comment applies to the recovery detection years, where the current results need to be put more into the context of existing literature. My Figs. 2 and 3 below, for example, compare recovery trend magnitudes and detection years from this study (Figs. 2 and 3) with those from Figure 3 of Weatherhead et al. (2000). Weatherhead

et al., from their 2 D model, find trends that are only about half the size of trends in the current study, and also find detection times that are about twice as long as in the current study. In my Figs. 2 and 3 below, I have scaled the Weatherhead et al. results to account for that (see also Eq. 2 of Weatherhead et al. 2000). The comparison in my Fig. 3 indicates that the expected detection years in the current study are generally earlier than in the Weatherhead et al. study, particularly in the tropics, but also at Northern mid-latitudes. Clearly the magnitude of the expected trend plays a large role, especially when trends go to zero (tropics). I think this needs to be brought out much clearer in the current manuscript.

3.) What also needs to be discussed more is one of the main messages of Fig. 3 of the current manuscript. Essentially, this figure says that by 2017 significant ozone recovery should have been detected between $15°$ and $50°$ latitude and in both hemispheres. In reality, however, I don't think that is the case, e.g. Weber et al. 2017. So what is going wrong? Is the model too optimistic? Are the uncertainty bars too small? Are the observations too bad? Is the atmosphere not doing what it is supposed to? I think these questions need to be discussed more, and could really be key points of the paper. Just pointing out the large sample size of the simulations (e.g. page 8 lines 16, 17) is not enough. Certainly, to be meaningful, these results need to be translated into something that is observable in the real world.

4.) In this general context, I am surprised about the small uncertainty of the detection years in the tropics in the authors' Fig. 3. Since the uncertainty of the trends in the authors' Fig. 2 includes zero, no trend is a possiblity, and detection of a significant trend would take forever. Why is that not reflected in the small tropical error bars in Fig. 3? Compare also the (much more realistic) large spread between the blue and red data points in the tropics in Fig. 4, or the late tropical detection years in Weatherhead et al. (2000).

5.) Generally, I am missing clear explanations, how the error bars where obtained in Figs. 2 to 4. See my detailed comments below for specifics.

6.) Since the authors have not really presented much information about point (i) the slowing of past ozone decline and the date of minimum column ozone, I suggest to delete this specific point, especially in abstract and conclusions. I agree with the authors statements in Section 4, especially page 5, lines 19, 20: the date of minimum ozone is a poor metric and therefore point (i) should not really be given much attention and should not be mentioned in abstract and conclusions.

**3  Detailed comments**

Page 1, lines 10, 11: See my major comment 6, above.

Page 1, lines 12 to 14: I find this sentence weird and confusing. Of course, all kinds of mistakes can be made. Maybe just drop this sentence, move the "(e.g. solar cycle, QBO, ENSO)" after "natural cycles" in the following sentence, and start that sentence with something like "Our investigations point to the need ..."

Page 1, line 17: See my major comment 3, above.

Page 1, line 18: What do you mean by "sizeable"? I think what you really mean is something like the ratio of trend to natural cycle variability, or trend to unexplained variability. Please reword, clarify.

Page 1, line 22: This is a good statement, but it is in conflict with the small tropical uncertainty bars in Figure 3. See also my major comments 4 and 5.

Page 1, line 25, "were shown to"; page 5, line 16, "seen to be"; page 7, line 9, "are found to": I suggest to drop such unnecessary wordings, possibly also in other places.

Page 2, line 5: Drop "the difference"?

Page 2, line 9: Drop "Solomon et al. 2016". I don't think that paper says much about changing BDC / ozone transports.

Page 2, line 20: I agree that the linear assumption is "somewhat simplistic". However, so many studies, including complex CCM studies, have shown that, in the end, the whole system behaves remarkably linear, and that the linear assumption does work very well. So maybe replace "somewhat simplistic" by "surprisingly robust"?

Page 2, line 33: Maybe drop that line, and reword the previous sentence? See my major comment 6.

Page 2, line 34: I would add "after accounting for natural variability" after "values". In fact you say and show later that accounting for these cycles is important. Obviously, unaccounted for ups and downs are prone to misinterpretation. So here, and in other places, little text and attention should be given to those "raw" results.

Page 2, lines 38, 39: I would drop "as proxies for atmospheric observations".

Page 3, line 24: I wonder about the sea-surface and ice conditions. For two reasons: In Fig. 1, the red line appears to be much smoother after 2000, and much more variable from 1960 to 2000. Are your runs using observed sea surface conditions before 2000, and some climatology after 2000? Do missing real surface conditions have something to do with the mismatch between your simulated trends and observed trends e.g. from Weber et al. (2017), see also my supplemented Fig. 1. I think you should clarify this, and also make some statements about the importance of sea surface conditions for these ozone trends. I think there is past work by Braesicke and others on the influence of sea surface conditions on the stratosphere, and probably a lot more to be cited here – ask John Pyle.

Page 3, lines 35, 36: This equation needs a lot more explanation. Are you using monthly means, or what? Are the data deseasonalized?

If the $TO3_{e,l,i}$ is to be meaningful ozone, all the predictors have to be normalized to mean 0, or to 0 under "normal" conditions. This should be stated.

What does the subscript $i$ mean in $TO3_{e,l,i}$? Calendar month?

Certainly the $a^x$ need to depend on latitude and be $a_l^x$.

Is the regression applied to all ensemble runs at once (providing $a_l^x$), or individually to each run (providing $a_{e,l}^x$)?

How do you deal with autocorrelation in the $N_{e,l,t}$? Autocorrelation can be substantial, e.g. $0.6$ in the tropics, (see Plate 3 of Weatherhead et al. 2000). This reduces effective sample size and increases error bars, and needs to be accounted for. In the same direction: How independent are the ensemble runs? In the model world they may be independent, but compared to the real world, they are not really independent samples drawn from a large population.

Page 4, lines 12 to 14: I would move that sentence much closer to the Equation. I think it is important to understand what is fitted.

Page 4, line 17: I am missing an explanation how the trends in Fig. 2 are obtained. Presumably for the MLR trends, you fit straight lines to the $RO3_{e,l,t}$ and obtain the trend uncertainty from the fit residuals / remaining noise? Again: Are all ensembles fitted at the same time, or do you fit each ensemble separatately?

Don't the $a^x$ need subscripts $e, l$? How is autocorrelation in the fit residuals dealt with?

How do you obtain the raw model trend in Fig. 2? By simply fitting straight lines to the $TO3_{e,l,t}$ on the left side of the first equation? Or piecewise linear trends? Please add text here or later, and answer these questions.

Page 5, lines 18, 19: See major comment 6. Same paragraph: I think you should also add some arguments based on your Fig. 1, e.g. the large uncertainty range for minimum ozone from 1992 to almost 2010, with little difference between blue and red curves.

Page 5, line 35: 95% confidence intervals – obtained how and from what? Please explain.

Page 5 lines 36, 37: "heterogeneous . . . vortex". Not only that. The Brewer Dobson Circulation also "transports" the large ozone trends from the upper stratosphere polewards and downwards into the lower stratosphere. There, near the ozone maximum, they make a big difference for column ozone (whereas otherwise upper stratospheric ozone does not contribute a lot to the total column). Please reword, or add some text.

Page 5, lines 38, 39: Please add "declining" or "1980 to 1997" before "trends". Otherwise this is misleading and might be mistaken with the increasing 2000 to 2017 trends.

At some point, you might also want to point out that by picking 1980 to 1997 and 2000 to 2017, you have picked two one-and-a-half solar cycle long periods. This would maximize solar cycle effects on the trends (e.g. solar max at one end, solar mind at the other end). So some of your results might include large solar cycle effects – but still the comparison of raw and MLR trends in Fig. 2 does not look too bad. You do have a corresponding discussion on page 6, lines 9 to 18. However that discussion reads a bit awkward, and, to me, puts too much focus on the "raw trends", which obviously are influenced by the solar cycle and obviously should not be used. Maybe reword that discussion.

Page 6, line 2: maybe add "and Pinatubo aerosol effects"

Page 6, line 10: add "and autocorrelation" after "variance" and "(Weatherhead et al. 2000)" after "data".

Page 6, line 19: Replace "month" by "year"? Also in other places in this paragraph?

Page 6, lines 19 to 30: How did you obtain the error bars in Fig. 3? From comparing results of the different runs? Is that realistic? See my major comments 4, 5.

Page 7, line 24: add "calendar" after "to", and replace "certain months" by "e.g., in September"

Page 7, line 35: same as major comment 6.

Page 8, lines 2 to 5: same as page 1, lines 12 to 14.

Page 8, lines 12 to 19: What about transport variations? You are not talking / accounting for them at all. See also my comment above about sea surface conditions. I think you should add something here, and also discuss the differences to the observations more, e.g. citing Weber et al. 2017 and Ball et al. 2017. See also my comment above about sample size, and my major comments 2 and 3.

Page 8, line 20: Same as Page 1, line 18.

Page 8, line 23: To me, it is worrying that precisely there Weber et al. 2017 find small and non-significant increases (see also Ball et al. 2017). I think you need to comment more on that, and I think this difference could be a key message from this study. See also my major comment 1.

Page 15, Figure 2: In the legend in the Figure. Please replace "Model trend" by "simple trend" or "raw trend". That would be clearer, and the "MLR" trends are "model" trends as well. In the caption, please explain how the error bars where obtained.

Page 16, Figure 3: In the caption, please explain how the error bars where obtained. See also my major comment 4.

Page 17, Figure 4: Why was that not done for the MLR / residual total ozone as well? Should that not be shown? In the caption, please explain how the error bars where obtained.

**4  References**

Weatherhead, E. C., Reinsel, G. C., Tiao, G. C., Jackman, C. H., Bishop, L., Frith, S. M. H., DeLuisi, J., Keller, T., Oltmans, S. J., Fleming, E. L., Wuebbles, D. J., Kerr, J. B., Miller, A. J., Herman, J., McPeters, R., Nagatani, R. M., and Frederick, J. E., 2000.

[Figure]

Detecting the recovery of total column ozone, J. Geophys. Res., 105, 22201-22210, https://doi.org/10.1029/2000JD900063.

Weber, M., Coldewey-Egbers, M., Fioletov, V. E., Frith, S. M., Wild, J. D., Burrows, J. P., Long, C. S., and Loyola, D., 2017. Total ozone trends from 1979 to 2016 derived from five merged observational datasets – the emergence into ozone recovery, Atmos. Chem. Phys. Discuss., https://doi.org/10.5194/acp-2017-853, in review.
* * *
[Figure]

[Figure]

Legend:
- Model trend 1980-1997
- MLR trend 1980-1997
- Model trend 2000-2017
- MLR trend 2000-2017

**Fig. 1.** Comparison of Fig. 2 of reviewed manuscript (trends in DU / year) with observed trends (% / decade) from Weber et al. 2017.

[Figure]

**Fig. 2.** Comparison of total ozone trends from the reviewed manuscript with Weatherhead et al. 2000 (black curve), scaled up by factor 2.

[Figure]

**Fig. 3.** Comparison of year for detection of significant increase from the reviewed manuscript with Weatherhead et al. 2000 (black curve), scaled down by factor of 2.

---

## Referee Comment (RC2) · Anonymous Referee #2 · 29 Jan 2018

**(1) General comments**:

The expected stratospheric ozone recovery from the effect of halogenated ozone depleting substances (ODSs) has received much attention in recent years. Yet detecting the recovery of the ozone layer is complex due to a number of factors, including internal and external variability, that obscure the emerging signal associated with the slow decline in ODSs levels. The manuscript addresses this issue by investigating three stages of ozone recovery. To this end, the authors use total column ozone (TCO) changes based on experiments of the UM-UKCA and multiple linear regression (MLR) analysis. Although models are not perfect (e.g. often show significant disagreement compared to observations), they are a valuable mean to explore ozone changes due to specific factors (i.e. ODSs levels).

Overall, the manuscript addresses relevant issues with regard to the evolution of the stratospheric ozone layer and uses appropriate data and methods. The text is technically well written. I have minor specific comments (detailed below), which I hope will help the authors improve the paper. In general, I suggest more detailed description and evaluation, additional comparison with ozone measurements, and further discussion on existing literature. Therefore, the manuscript is recommended for publication after the specific and technical comments are addressed.

**(2) Specific comments**:

**a**. In the Introduction section, the authors clearly set out the stratospheric ozone depletion in the last decades associated with man-made emissions of ODSs. Due to international efforts banning the use of these substances, the ozone layer is expected to recover and the study aims to explore different stages. However, significant work has been done on detection and attribution of ozone recovery, hence it would be appropriate (and helpful for the broader audience) to briefly introduce key findings, remaining issues, and link it with the novelty of this work. Moreover, this will help relate and put into context the main findings here later in the manuscript.

**b**. In the Model configuration and simulations section (page 3, lines 25–26), the authors explain that the simulations used were performed in support of the CCMI activity, and that are described in more detail in Bednarz et al. (2016) and Keeble et al. (2017). Bednarz et al. (2016) described that the simulations included a future

climatological solar cycle since 2009 based on the observed cycle 23, which is not consistent with the description given in the manuscript (page 3, lines 16–17). Please clarify.

**c**. In the Removing natural cycles section, the text describes a MLR analysis to identify the impacts of natural variability on TCO. Since the results of this study heavily rely on the MLR analysis, I think this section requires more detailed description of the statistical method. In particular, the $TO3_i$ and $N_{e,l,t}$ terms need better description (i.e. "… some constant value" and "Any noise…"). Also, an evaluation of the MLR analysis is important – i.e. How good is it? How much of the model raw data is captured by the MLR and how much "noise" is left? –. The manuscript already includes some references on MLR analysis that may help.

**d**. For the Modelled global column ozone and minimum values section, it may be appropriate a statement about the choice of not including the polar regions in this analysis (Figure 1), since, in other sections and figures these regions are included and also discussed in the last paragraph here (page 5, lines 18–25). In fact, the latter paragraph argues that minimum column ozone values are a poor indicator of ozone recovery by giving examples based on polar regions.

Is there any particular reason for not using the latest version (3.3) of the Bodeker Scientific database? The latest version, in addition to include some improvements on the methodology, could be expanded until 2016 in the inset of Fig. 1. Also it would be nice to include the Bodeker Scientific database in the acknowledgements, as recommended on the website.

**e**. Regional trends section. This section includes very interesting results. However, modelled results in Fig. 2, both "raw" and residual data, could be compared to observed trends. In turn, this may lead to some evaluation/discussion and to put into context these results with existing literature. Nevertheless, there is some discussion (outlook) on TCO trends between 2000–2017 in the Discussion and Conclusions section (page 8, lines 15–19).

Error bars representing the 95% confidence interval may need a line or two detailing how these are estimated and whether they account for autocorrelation. Are these confidence intervals calculated in the same way for all analyses?

**f**. Return to historic values section. Figure 4 shows that TCO values in the tropics (<30º) reach the "1980 last recovery" between ~2060s–2070s. However, the main text (page 7, lines 12–14) explains that "…, it is the only region in which total column ozone abundances are not greater than their 1980s values by the end of the simulation,…". Please clarify. Also, is there any particular reason for not showing(using) "ozone residuals" on Fig. 4 as in previous analyses? I understand the study aims to explore ozone recovery addressing natural cycles.

**(3) Technical comments**:

**Page 1, lines 9–10**. "This approach…". The approach or method has not really been introduced. I suggest rephrasing this sentence (e.g. Here internal atmospheric variability… is accounted for by…).

**Page 1, line 28**. Substitute "ODSs" for "ODS" for consistency throughout the text.

**Page 1, line 34**. Randel and Wu (1995) did not explore the effects of Mt Pinatubo eruption on stratospheric ozone.

**Page 1, lines 37–38**. References order.

**Page 1, line 38**. Delete "," between "other" and "non-chlorinated".

**Page 2, line 2**. References order.

**Page 2, lines 13–15**. "Good agreement…" This sentence is a bit confusing, rephrasing maybe?

**Page 2, lines 22–23**. "…, data from fully coupled chemistry-climate model…" is a bit misleading since you use imposed SSTs. I would clarify "fully coupled" (chemistry and radiation schemes?).

**Page 2, line 26**. Spell out "SSTs".

**Page 2, line 37**. Substitute "-" from "–".

**Page3, line 23**. Could use just "SSTs", as it was introduced before.

**Page 5, line 33**. Substitute "… DU year$^{-1}$…" for "… TCO (DU year$^{-1}$)…"? I am aware that "for the column ozone" is mentioned later in the sentence, though it is somehow confusing.

**Page 7, lines 9–10**. "… 1980s values for the first time (light red)…" should be "blue".

**Page 7, line 21**. Typo: "airmasses".

**Page 7, line 26**. I would substitute "expected" for "projected" (e.g. acknowledging these are modelled results, which are model and scenario dependent).

**Page 8, line 15**. Typo: "… Unlike a recent analyses…"

**Page 8, line 26**. Typo: "… The tropics have too small a trend…"

**Page 15, line 2; and Figure 2, legend**. Please follow same consistency in the naming, both for the figure and the main text.

---

## Author Comment (AC1) · 27 Apr 2018

Response to Anonymous Referee #1

General comments:

Based on simulations by the UM-UKCA chemistry climate model, the authors investigate past and current trends in total column ozone, the years when the expected increases in total ozone might become significant, and the years when total ozone might return to 1980s levels. These questions are relevant for the expected recovery of the ozone layer, and for checking the success of the international Montreal Protocol protecting the ozone layer. The used data and methods appear solid. The paper is generally clear, concise and well written. What I am missing, however, is a more thorough comparison with observed trends, and with existing literature. I am also missing a more detailed explanation, how uncertainties were estimated. I think all this can be fixed in a revised version. After addressing my comments below, the paper should be acceptable for publication in ACP.

**We thank the referee for their positive and detailed comments. Our detailed response is given below in bold**

Major comments

1.) I think it is absolutely necessary to compare the simulated trends and their uncertainties with trends from observations. In my Fig. 1 below, I have overlaid observed trends from Fig. 7 of Weber et al. (2017) onto the simulated trends from Fig. 2 of the current manuscript. Note that the trends from Weber et al. are in % per decade, so at higher latitudes they tend to appear smaller compared to the DU per year trends of the current manuscript. Nevertheless, the observed trends appear to be much smaller at mid-latitudes and in both hemispheres. The uncertainties, on the other hand seem to be quite comparable. A comparison like that would be extremely valuable. I urge the authors to add such a comparison to their paper. Preferably this would be in an additional Figure, comparing simulated and observed trends (e.g. from Weber et al. 2017), as well as their uncertainties, and using the same units (DU per time or % per time). If this comparison confirms the impression from my Fig. 1 below, and the observed trends at mid-latitudes are indeed much smaller, this would be an important finding. Such an apparent lack of significant ozone increases at mid-latitudes would question our expectations for ozone recovery (see also Ball et al. 2017).

**We have added reference to recent studies (e.g. Pawson et al., 2014; Weber et al., 2018) throughout the manuscript. Discrepancies between observed and modelled trends in the mid latitudes is indeed an important concern, as highlighted by the reviewer, and one we explore in the revised version of the manuscript. Comparison of modelled and observed trends is, of**

course, very complicated. The reviewer has overlaid our modelled trends (in DU/yr) with the trends calculated by Weber et al. (2018) from the NASA dataset (in %/decade). One should note the difference in calculated trends between the datasets presented in Fig 7 of Weber et al. Trends in the NASA dataset are generally smaller and less statistically significant than those of, for example, the WOUDC and GSG datasets. Further, Weber et al. state that the trends they calculate are roughly half of those presented by Pawson et al. in the latest ozone assessment (although still within the uncertainties of these trends), and attribute this to low total column ozone values at the end of their datasets. These low ozone values are stated to be within the expected interanual variability, and therefore part of the system noise rather than reflect a change in trend/processes. While our figure 1 shows that the model in general accurately reflects the observed interannual variability (e.g. by comparison with the Bodeker dataset), there is no reason to expect a free running model to reflect individual anomalies in the noise (particularly when using ensembles where these features would be averaged out between high and low ensemble members when calculating the mean trend) Therefore our trends are likely greater than those of Weber et al. as we do not model these low ozone years at the end of the record. We feel that while this does not reflect some shortcoming of the model, or some discrepancy of total column ozone trends, it could equally well result from the inability of a model projection to accurately predict values of total column ozone for individual years even if the projected trend and interannual variability are well captured.

Converting our modelled trends into %/decade, we get the following trends between 60S-60N:

| latitude | -60 | -50 | -40 | -30 | -20 | -10 | 0 | 10 | 20 | 30 | 40 | 50 |
|---|---|---|---|---|---|---|---|---|---|---|---|---|
| DU/yr | 0.67 | 0.45 | 0.30 | 0.18 | 0.12 | 0.09 | 0.10 | 0.17 | 0.27 | 0.42 | 0.56 | 0.62 |
| %/dec | 1.98 | 1.39 | 1.01 | 0.65 | 0.45 | 0.34 | 0.37 | 0.62 | 0.94 | 1.35 | 1.63 | 1.68 |

These trends are in general larger than those presented by Weber et al., but consistent with those of Chehade et al. (2012) and Pawson et al. (2014) within trend uncertainties of both studies. This leads us to conclude, as Weber et al. also concluded, that the smaller trends presented by their study in comparison to earlier observations studies and now to our model study is the result of recent years with low annual means compared to previous years that result from interannual variability. While it is possible that some process that is not captured by the model is responsible for the discrepancy between our modelled trends and those of Weber et al. (most likely transport related, e.g. Ball et al., 2017), there are not enough years of observation to state definitively that this discrepancy does not relate to interannual variability and low column ozone values at the end of the observation record not present in the study of Pawson et al.

**This discussion has been added to the manuscript in section 5 (discussion of regional trends) and section 7 (discussion and conclusions). Please also note that direct comparison of our model data with the results of Weber et al is further complicated by differences in the MLR analysis performed in the two studies and the need when comparing modelled and observed quantities to compare collocated data points. As such, we feel it is beyond the scope of this paper (and probably misleading) to produce a figure with both datasets on and to undertake the analysis this properly requires. We do however take the reviewer's point and, as requested, significant changes have been made to the manuscript to discuss the comparison of our data with the studies of Weber et al., and Pawson et al.**

2.) A similar comment applies to the recovery detection years, where the current results need to be put more into the context of existing literature. My Figs. 2 and 3 below, for example, compare recovery trend magnitudes and detection years from this study (Figs. 2 and 3) with those from Figure 3 of Weatherhead et al. (2000). Weatherhead et al., from their 2 D model, find trends that are only about half the size of trends in the current study, and also find detection times that are about twice as long as in the current study. In my Figs. 2 and 3 below, I have scaled the Weatherhead et al. results to account for that (see also Eq. 2 of Weatherhead et al. 2000). The comparison in my Fig. 3 indicates that the expected detection years in the current study are generally earlier than in the Weatherhead et al. study, particularly in the tropics, but also at Northern mid-latitudes. Clearly the magnitude of the expected trend plays a large role, especially when trends go to zero (tropics). I think this needs to be brought out much clearer in the current manuscript.

**Total column ozone trends present here are around a factor of 2 larger than those of Weatherhead et al. (2000). This most likely is a result of the inclusion of CO2 induced cooling of the stratosphere in our simulations (and the resulting increases in O3), while the simulation performed by Weatherhead et al. using the GSFC 2-D model "assumes no direct temperature changes due to greenhouse gas emissions." Many studies (e.g. from Haigh and Pyle, 1982 onwards and including recent studies by Fleming et al., 2011; WMO, 2014; Butler et al., 2016, Keeble et al., 2017) have shown that CO2 induced cooling of the stratosphere contributes significantly to projected future O3 increases at a range of latitudes. Accounting for this difference, we feel that the key finding is consistent between the two studies – significant recovery trends will first be identified in the mid latitudes due to the relatively large trends and small interannual variability in these regions. There is an offset in when trends are expected to become significant, with our modelled projected detection years earlier than those of Weatherhead et al., particularly in the tropics. This is unsurprising, as tropical column ozone recovery is dependent on declining CFCs, increasing GHG and changing BDC speeds (see e.g. Eyring et al., 2013; Meul et al., 2016; Keeble et al., 2017), with increases to the BDC offsetting**

ozone recovery in the lower stratosphere and resulting in smaller recovery trends. However, the major cause for this difference likely comes from the large number of data points when using the ensemble of 7 simulations to determine significance. We calculate the trend magnitude from January 2000 to month x for each ensemble member, and then increment x by one until trends have been calculated for all months from 2000 to 2080. Trend significance is calculated using data from all 7 ensemble members so that it is comparable with trend significances calculated for figure 2. Then an array of months is constructed for trends which are positive and significant at the 95% confidence level. Initial recovery is defined as the first time a significant trend can be identified (excluding the first 36 months (i.e. up to January 2003) to prevent identifying significant trends when too few data points are present). The error bars on our figure 3 represent the 95% confidence interval (~2sigma) of the range of these 7 values. In this way we explore the impacts of the unexplained noise term on trend detection. In comparison, Weatherhead et al. calculate the trend and trend uncertainty for modelled column ozone values for the years 2000-2020 from a single model integration, and then use these terms in their equation 2 to calculate the number of months required. As a result there are likely large differences in the trend significance term, and hence detection dates. It should be noted that no individual ensemble member shows significant trends at any latitude by 2017 when trend significance is determined using data from only one ensemble member.

The discussion of figure 3 has been modified to reflect these points (see section 5: regional trends), and at the end of the section the following text has been added comparing our results with the Weatherhead results:

"Once the difference in projected recovery trends is accounted for, these findings are consistent with those of Weatherhead et al. (2000), who also identified the midlatitudes as the best location to identify early signs of ozone recovery. However, there is an offset in when trends are expected to become significant between the two studies, with projected detection years modelled in this study generally occurring earlier than those of Weatherhead et al. (2000), particularly in the tropics. This discrepancy in the tropics is unsurprising, as recovery of the tropical ozone column is dependent on the competing influences of declining CFCs, decreasing stratospheric temperatures and changing BDC speeds (see e.g. Eyring et al., 2013; Meul et al., 2016; Keeble et al., 2017), with increases to the BDC offsetting ozone recovery in the lower stratosphere and resulting in smaller column ozone recovery trends. There is poor agreement in modelled projections of future BDC speeds, and as a result projections of tropical column ozone differ significantly between models (e.g. WMO, 2011)."

3.) What also needs to be discussed more is one of the main messages of Fig. 3 of the current manuscript. Essentially, this figure says that by 2017 significant ozone recovery should have been detected between 15_ and 50_ latitude and in both hemispheres. In reality, however, I don't think that is the case, e.g. Weber et al. 2017. So what is going wrong? Is the model too optimistic? Are the uncertainty bars too small? Are the observations too bad? Is the atmosphere not doing what it is supposed to? I think these questions need to be discussed more, and could really be key points of the paper. Just pointing out the large sample size of the simulations (e.g. page 8 lines 16, 17) is not enough. Certainly, to be meaningful, these results need to be translated into something that is observable in the real world.

**The early detection of trends in figure 3 is attributable to the large number of points available when using and ensemble of integrations. For figure 3, trend significance is calculated using the trend uncertainty obtained when all ensemble members are used but the trend magnitude is calculated for each ensemble member individually, to provide a range of dates that trend significance can be identified. This range of dates reflects the impact of unaccounted for noise on the trend magnitude, while maintaining significance thresholds which are consistent between figures 2 and 3. It should be noted that no individual ensemble member shows statistically significant recovery by 2017, and only when considering data from all the ensembles together when calculating trends uncertainties can significant trends be identified in the mid latitudes. This can most easily be seen when considering the trends shown for figure 2. Trend uncertainties are calculated using the equation from Weatherhead et al. (1998), in which the standard deviation of the uncertainty in the linear trend is calculated by:**

$$\sigma_{trend} = \frac{\sigma_{data}}{n^{3/2}} \sqrt{\frac{1 + \varphi}{1 - \varphi}}$$

**When using all ensemble members together, n=17x12x7=1428, while for each individual ensemble member n=17x12=204, while the $\sigma_{data}$ term is roughly comparable between each ensemble member and when the full ensemble is used. As a result, trend significance can be determined earlier than when using observations/single member CCM integrations.**

4.) In this general context, I am surprised about the small uncertainty of the detection years in the tropics in the authors' Fig. 3. Since the uncertainty of the trends in the authors' Fig. 2 includes zero, no trend is a possiblity, and detection of a significant trend would take forever. Why is that not reflected in the small tropical error bars in Fig. 3? Compare also the (much more realistic) large spread between the blue and red data points in the tropics in Fig. 4, or the late tropical detection years in Weatherhead et al. (2000).

**In our response to the points above we have detailed how we calculate trend uncertainty and identify significant recovery trends. The small uncertainty in trend detection results primarily from the use of a large ensemble of model runs, which reduces the trend uncertainty resulting from interannual variability not accounted for in the MLR. As discussed above, we use data from all 7 ensemble members to calculate trend uncertainties, and the spread in dates obtained for figure 3 results only from the difference in trend magnitude between each ensemble member, which is not significantly different in the tropics as interannual variability is low here in comparison to the high latitudes. Interpretation of these trends in the tropics is hard as the change in column ozone resulting from CFC changes is small in comparison to interannual variability. Further, recovery trends in the tropics are also driven by $CO_2$ induced cooling of the upper stratosphere and changes to the BDC (see, e.g., Keeble et al., 2017 who analyse partial column ozone changes in this ensemble). These factors contribute at least as strongly to column ozone trends as chlorine catalysed ozone depletion.**

5.) Generally, I am missing clear explanations, how the error bars where obtained in Figs. 2 to 4. See my detailed comments below for specifics

**We have endeavoured to add more detail about how error bars are calculated for all figures. Please see our responses to the detailed comments below.**

6.) Since the authors have not really presented much information about point (i) the slowing of past ozone decline and the date of minimum column ozone, I suggest to delete this specific point, especially in abstract and conclusions. I agree with the authors statements in Section 4, especially page 5, lines 19, 20: the date of minimum ozone is a poor metric and therefore point (i) should not really be given much attention and should not be mentioned in abstract and conclusions.

**We feel that this point should remain in the abstract and conclusions as, firstly, identification of the minimum values is the earliest indication that ozone has stopped declining, and secondly that the timing of minimum values is a further metric by which to assess the models performance. There is good agreement between the simulations presented here and the timing of the minimum ozone values in the Bodeker dataset. These points are addressed in section 4, although briefly because, as the review states, this metric is a poor one for determining anything quantitatively about recovery, and as a result we feel should also be present in the abstract and conclusion.**

Detailed comments

Page 1, lines 10, 11: See my major comment 6, above.

**Please see discussion above**

Page 1, lines 12 to 14: I find this sentence weird and confusing. Of course, all kinds of mistakes can be made. Maybe just drop this sentence, move the "(e.g. solar cycle, QBO, ENSO)" after "natural cycles" in the following sentence, and start that sentence with something like "Our investigations point to the need . . ."

**This sentence is intended to reflect the important point that while a recovery trend may be deemed statistically significant for some set of years, when additional years are included the recovery trend may become non-significant. From our data presented in Figure 3 this effect is seen for the recovery trend at almost all latitudes, but is clearest at 60-70N where the trend from 2000-2005 is statistically significant at the 95% confidence level, but the trend from 2000-2010 is not. It is not until ~2014 that a statistically significant trend is modelled that remains significant no matter how many additional years are considered. This effect is arising from interannual variability. Separating initial and robust recovery in this way is important as while it can be calculated easily using model projections, it cannot be known for observed trends as we do not know what the future observed values will be. Therefore, we feel it is an important point to remain in the abstract, but have amended the sentence to prevent confusion. It now reads:**

**"It is important to note that, while a statistically significant recovery trend could be calculated at a particular point of time, additional years of observations may lead to a reduced significance of trends due to either a decrease in the magnitude of the trend or an increase in interannual variability. This highlights the need to ensure that the impact of natural cycles (e.g. solar cycle, QBO, ENSO) on total ozone is correctly described in statistical models, especially in the tropics where chemical depletion of the column is small."**

Page 1, line 17: See my major comment 3, above.

**More detail has been added to the manuscript regarding the calculation of trends and their uncertainties in this study and the implications of the results presented here. Please see our response to comment 3 above for detail.**

Page 1, line 18: What do you mean by "sizeable"? I think what you really mean is something like the ratio of trend to natural cycle variability, or trend to unexplained variability. Please reword, clarify.

**This sentence has been amended to read "The influence of the natural cycles on trend determination is least at latitudes where the trends are sizeable and the ratio of trend magnitude to interannual variability is large."**

Page 1, line 22: This is a good statement, but it is in conflict with the small tropical uncertainty bars in Figure 3. See also my major comments 4 and 5.

**This sentence has been amended to better reflect the findings of this study. It now reads: "Significant trends cannot be identified by 2017 at the highest latitudes, due to the large interannual variability in the data, nor in the tropics, due to the small trend magnitude, although it is projected that significant trends may be identified in these regions soon thereafter."**

Page 1, line 25, "were shown to"; page 5, line 16, "seen to be"; page 7, line 9, "are found to": I suggest to drop such unnecessary wordings, possibly also in other places.

**These phrases have been removed from the text**

Page 2, line 5: Drop "the difference"?

**This 'difference' is here to make clear that interannual variability varies with latitude, and we feel it should remain.**

Page 2, line 9: Drop "Solomon et al. 2016". I don't think that paper says much about changing BDC / ozone transports.

**This reference has been removed from the discussion of BDC/transport changes.**

Page 2, line 20: I agree that the linear assumption is "somewhat simplistic". However, so many studies, including complex CCM studies, have shown that, in the end, the whole system behaves remarkably linear, and that the linear assumption does work very well. So maybe replace "somewhat simplistic" by "surprisingly robust"?

**'somewhat simplistic' has been removed so that the sentence now reads "These statistical approaches nearly all work by relying on the assumption of a linear relationship between a proxy variable and its impact on total ozone."**

Page 2, line 33: Maybe drop that line, and reword the previous sentence? See my

major comment 6.

**Please see discussion relevant to major comment 6.**

Page 2, line 34: I would add "after accounting for natural variability" after "values". In fact you say and show later that accounting for these cycles is important. Obviously, unaccounted for ups and downs are prone to misinterpretation. So here, and in other places, little text and attention should be given to those "raw" results.

**This has been added to the manuscript**

Page 2, lines 38, 39: I would drop "as proxies for atmospheric observations".

**This phrase has been removed**

Page 3, line 24: I wonder about the sea-surface and ice conditions. For two reasons: In Fig. 1, the red line appears to be much smoother after 2000, and much more variable from 1960 to 2000. Are your runs using observed sea surface conditions before 2000, and some climatology after 2000? Do missing real surface conditions have something to do with the mismatch between your simulated trends and observed trends e.g. from Weber et al. (2017), see also my supplemented Fig. 1. I think you should clarify this, and also make some statements about the importance of sea surface conditions for these ozone trends. I think there is past work by Braesicke and others on the influence of sea surface conditions on the stratosphere, and probably a lot more to be cited here – ask John Pyle.

**SST and sea ice fields for the entire length of each simulation used in this study are taken from an integration preformed with another coupled ocean model (HadGEM2-ES), as discussed in section 2 (p3, l25). As a result, there is no abrupt change in this lower boundary condition, which would result from switching from observed to modelled SSTs. However, the result is that the modelled SSTs differ from observed SSTs, and as a result the timings of ENSO events also differs. While this will affect interannual variability for the raw model values, the MLR analysis includes an ENSO term and as a result this shouldn't influence the trends derived from the ozone residuals in a significant way. As a result, it is unlikely that differences between the trends of Weber et al. and those presented here result from differences in SSTs.**

**The fact that the red line appears to be much smoother after 2000, and much more variable from 1960 to 2000 most likely results from the prescription of stratospheric aerosols in the model. Historic stratospheric aerosol loadings are prescribed from 1979 through 2004**

**(following the SPARC dataset) and outside of these times background aerosol loadings are prescribed. As a result, historic volcanic eruptions (most notably El Chichon in '83 and Pinatubo in '91) are included. While there is a term in the MLR regression for stratospheric aerosol loadings, the effects on ozone of volcanic eruptions is dependent on stratospheric chlorine levels (as we discuss in section 7, p8 l14 and also discussed by Weber et al.). Therefore, the impacts of these eruptions are likely not completely removed, resulting in the relative difference in perceived interannual variability before and after 2000.**

**Braesicke and Pyle (2004) has been added as a reference to paragraph 1 of the introduction where SSTs are discussed.**

Page 3, lines 35, 36: This equation needs a lot more explanation. Are you using monthly means, or what? Are the data deseasonalized?

**We use deseasonalized monthly mean data. This point has been added to the manuscript in the paragraph following the first equation.**

If the $TO3_{e;l;i}$ is to be meaningful ozone, all the predictors have to be normalized to mean 0, or to 0 under "normal" conditions. This should be stated.

**Here $TO3_{e;l;I}$ does not correspond to any particular modelled data point, but instead corresponds to the intercept term of the MLR**

What does the subscript i mean in $TO3_{e;l;i}$? Calendar month?

**$TO3_i$ is a constant value of total column ozone, corresponding to the y intercept term of each MLR. To avoid confusion, this term has been replaced by $i_{e,l}$**

Certainly the ax need to depend on latitude and be axl. **AND** Is the regression applied to all ensemble runs at once (providing axl), or individually to each run (providing axe;l)?

**The MLR analysis is applied to each model run separately, resulting in the 7 light red lines in figure 1. Subscripts for the alpha terms have been added to the regression model**

How do you deal with autocorrelation in the $N_{e;l;t}$? Autocorrelation can be substantial, e.g. 0:6 in the tropics, (see Plate 3 of Weatherhead et al. 2000). This reduces effective sample size and increases error bars, and needs to be accounted for. In the same direction: How independent are the ensemble

runs? In the model world they may be independent, but compared to the real world, they are not really independent samples drawn from a large population.

**Autocorrelations is accounted for when calculating confidence intervals for trends using the methodology described Weatherhead et al., 1998, such that:**

**std.dev$_{trend}$ = std.dev$_{noise}$/n$^{3/2}$.sqrt((1+phi)/(1-phi))**

**The 95% confidence interval is then calculated as 2* std.dev$_{trend}$**

**This point has been added to the text at the end of section 3.**

**Regarding the ensemble, it is not accurate to assume that each model integration is independent. Each integration uses the same boundary forcings, differing only in their initial conditions. As such, while this provides some information on projection uncertainty, it likely does not fully capture it, and as a result uncertainty estimates may be a minimum.**

Page 4, lines 12 to 14: I would move that sentence much closer to the Equation. I think it is important to understand what is fitted.

**This sentence has been moved towards the start of the paragraph, so that it now reads:**
**"in which the α values are the coefficients returned from the MLR for each explanatory variable (denoted by the superscript) and vary between latitude range and ensemble member. The explanatory variables included in the MLR are the QBO, solar cycle, ENSO, volcanic aerosols and ESC. The subscripts *e*, *l* and *t* indicate that the alpha value or explanatory variable differs with ensemble member, latitude and/or time respectively. For the QBO, two terms are included, *QBO$_{50}$* and *QBO$_{30}$*, which correspond to equatorial westerly winds at 50 hPa and 30 hPa respectively. Two QBO terms are included to account for the phase shift in the total column ozone response with respect to QBO changes at different altitudes. The solar cycle is represented by the top of atmosphere solar flux, represented in the MLR as *solar$_t$*. ENSO effects on column ozone are represented by *ENSO$_t$*, the detrended sea surface temperature anomalies in the NINO3.4 region. Volcanic aerosols are included as hemispheric aerosol optical depths, and so are different for the northern and southern hemispheres to account for the lack of interhemispheric transport of aerosols emitted into the stratosphere from high latitude eruptions. The final term included in the MLR, *ESC*, represents stratospheric chlorine concentrations. This term is equal to the ESC concentration at 30 km for each latitude bin to account for the time taken for ODS to be transported to higher latitudes. Any month to month**

variation not accounted for by the explanatory variables in the MLR is represented by the noise term $N_{e,l,t}$. The *Solar$_t$*, *ENSO$_t$* and *aerosol$_t$* terms are all prescribed forcings in the model and do not vary between ensemble members. In this study we use deseasonalised monthly mean total column data, and so there is no need for a seasonal cycle term in the MLR model."

Page 4, line 17: I am missing an explanation how the trends in Fig. 2 are obtained. Presumably for the MLR trends, you fit straight lines to the RO3e;l;t and obtain the trend uncertainty from the fit residuals / remaining noise? Again: Are all ensembles fitted at the same time, or do you fit each ensemble separatately?

**Trends for the raw model data and MLR data are obtained by fitting independent linear trends to the modelled total column ozone values (TCO3 on LHS of first equation) and the ozone residuals produced when accounting for natural cycles (RO3 on LHS of second equation). All ensembles are fitted at the same time. This information has been added to the manuscript at the end of section 3 by adding the following paragraph:**

**MLR analysis and ozone residuals are produced for each individual ensemble member of the raw model data, resulting in seven RO3 time series. For both the raw model data and the ozone residuals, decline and recovery trends are calculated using a independent linear trend fit for the periods 1980-1997 and 2000-2017 respectively. When calculating trends for both the raw model data and ozone residuals, a single linear fit is produced using data from all seven ensemble members rather than producing a fit for each individual ensemble. In order to make any robust conclusions about the statistical significance of modelled trends, some measure of the trend uncertainty is required. Here we calculate trend uncertainties following the methodology of Weatherhead et al. (1998), in which the standard deviation of the uncertainty in the linear trend is calculated by:**

$$\sigma_{trend} = \frac{\sigma_{data}}{n^{3/2}} \sqrt{\frac{1 + \varphi}{1 - \varphi}}$$

**where $\sigma_{data}$ is the standard deviation of the time series in question (either raw model data or RO3), $n$ is the number of months in the time series and $\varphi$ is the autocorrelation coefficient for a 1 month lag. As discussed by Weatherhead et al. (1998) autocorrelation can be substantial for monthly mean time series, particularly in low latitudes, and failure to account for it results in an underrepresentation of trend uncertainty. As for the trend calculations, trend uncertainties are calculated using data from all seven ensemble members together, and as such for the 17 year periods considered for the decline and recovery phases $n$ is very large ($n=17\text{x}12\text{x}7=1428$).**

Don't the ax need subscripts e; l? How is autocorrelation in the fit residuals dealt with? How do you obtain the raw model trend in Fig. 2? By simply fitting straight lines to the TO3e;l;t on the left side of the first equation? Or piecewise linear trends? Please add text here or later, and answer these questions.

**Alpha terms do need e,l subscripts, and these have been added to both equations. Autocorrelation is dealt with following Weatherhead et al. (1998), as detailed above. Independent linear trends are used to obtain trends from the raw model data. This information ha been included in the manuscript at the end of section 3 in the text provided for the comment above.**

Page 5, lines 18, 19: See major comment 6. Same paragraph: I think you should also add some arguments based on your Fig. 1, e.g. the large uncertainty range for minimum ozone from 1992 to almost 2010, with little difference between blue and red curves.

**The following sentence has been added to the last paragraph of section 4:**

**"Even outside the high latitudes, where interannual variability in total column ozone values is largest, identification of the year of minimum ozone is uncertain, with each of the seven residual ozone time series having minimum values at different times between 1992-2005 (light red lines, Figure 1)."**

Page 5, line 35: 95% confidence intervals – obtained how and from what? Please explain.

**Information on how the trend uncertainties are calculated has been added to the end of section 3, and in section 5 we stipulate that the 95% confidence intervals = $2\sigma_{trend}$**

Page 5 lines 36, 37: "heterogeneous . . . vortex". Not only that. The Brewer Dobson Circulation also "transports" the large ozone trends from the upper stratosphere polewards and downwards into the lower stratosphere. There, near the ozone maximum, they make a big difference for column ozone (whereas otherwise upper stratospheric ozone does not contribute a lot to the total column). Please reword, or add some text.

**This sentence has been expanded to include reference to the effects of BDC transport of ozone anomalies, so that it now reads:**

**"During the decline phase, ozone trends from both datasets are greatest at high latitudes due to the heterogeneous activation of chlorine on PSCs within the polar vortex and the transport of midlatitude polar ozone depletion signals to high latitudes by the Brewer-Dobson Circulation."**

Page 5, lines 38, 39: Please add "declining" or "1980 to 1997" before "trends". Otherwise this is misleading and might be mistaken with the increasing 2000 to 2017 trends.

**To avoid confusion, this sentence has been changed to:**
**"Negative trends in the raw column ozone data from 1980-1997 are significant at all latitudes…"**

At some point, you might also want to point out that by picking 1980 to 1997 and 2000 to 2017, you have picked two one-and-a-half solar cycle long periods. This would maximize solar cycle effects on the trends (e.g. solar max at one end, solar mind at the other end). So some of your results might include large solar cycle effects – but still the comparison of raw and MLR trends in Fig. 2 does not look too bad. You do have a corresponding discussion on page 6, lines 9 to 18. However that discussion reads a bit awkward, and, to me, puts too much focus on the "raw trends", which obviously are influenced by the solar cycle and obviously should not be used. Maybe reword that discussion.

**This point has been raised when discussing, in section 5, the difference in trend magnitude for the raw and ozone residual trends in figure 2. This discussion has been reworded to clarify the point above and avoid confusion.**

Page 6, line 2: maybe add "and Pinatubo aerosol effects"

**This sentence has been changed to read:**
**"This is the result of the eruption of Mt Pinatubo and the pronounced solar minima during the 1990s, both of which resulted in lower column ozone values and so a greater trend from 1980."**

Page 6, line 10: add "and autocorrelation" after "variance" and "(Weatherhead et al. 2000)" after "data".

**These have been added to the text**

Page 6, line 19: Replace "month" by "year"? Also in other places in this paragraph?

**The first instance of month has been replaced by year, as this is what is plotted in Fig 3. However, the script used to calculate trend significance here iterates over each successive month of ozone residual data, and so use of months within the text of the manuscript is more accurate than year. The first two sentences of this paragraph have been altered to reflect this.**

Page 6, lines 19 to 30: How did you obtain the error bars in Fig. 3? From comparing results of the different runs? Is that realistic? See my major comments 4, 5.

**Please see our response to the major comments above for details on how error bars and uncertainties are calculated for figure 3.**

Page 7, line 24: add "calendar" after "to", and replace "certain months" by "e.g., in September"

**This sentence has been changes to:**
**"However, the signature of this recovery is very sensitive to calendar month, and earlier signs of recovery may be identified in certain months (e.g. September; Solomon et al., 2016)"**

Page 7, line 35: same as major comment 6.

**Please see our response to major comment 6.**

Page 8, lines 2 to 5: same as page 1, lines 12 to 14.

**As with our response to the earlier comment, this sentence has be changes to avoid confusions. It now reads:**
**"However, an important caveat is that while a statistically significant recovery trend could be calculated at a particular point of time, additional years of observations may lead to a reduced significance of trend due to either a decrease in the magnitude of the trend or an increase in interannual variability. This highlights the need to ensure that the impacts of natural cycles (e.g. solar cycle, QBO, ENSO) on total ozone are correctly described in the MLR."**

Page 8, lines 12 to 19: What about transport variations? You are not talking / accounting for them at all. See also my comment above about sea surface conditions. I think you should add something here, and also discuss the differences to the observations more, e.g. citing Weber et al. 2017 and Ball et al. 2017. See also my comment above about sample size, and my major comments 2 and 3.

**Transport changes of course play a key role in short and long term ozone recovery trends. Better comparison with the observed trends of Weber et al. (2018) and Pawson et al. (2014) has**

**been added to the manuscript, as well as discussion of the transport changes discussed by Ball et al. (2018). We conclude that any transport changes which have affected the observed column and profile recovery are not captured by the ensemble mean and instead represent a feature of interannual variability and not some fundamental shift in stratospheric circulation driven by, for example, increased GHG concentrations.**

Page 8, line 20: Same as Page 1, line 18.

**This sentence has been clarified by changing it to:**
**"The influence of the natural cycles on trend determination is least at latitudes where the trends are sizeable and the ratio of trend magnitude to interannual variability is large."**

Page 8, line 23: To me, it is worrying that precisely there Weber et al. 2017 find small and non-significant increases (see also Ball et al. 2017). I think you need to comment more on that, and I think this difference could be a key message from this study. See also my major comment 1.

**This is a difficult point to evaluate as different datasets studied by Weber et al. give significant trends in different locations. In general, our results are consistent with the findings of fig 7 in Weber et al., who identify significant positive trends in the northern midlatitudes in some datasets (particularly WOUDC and GSG), although do not see such robust recovery in southern midlatitudes. This likely results from reduced trend uncertainty in our study due to the large number of datapoints provided by the ensemble members, and also the larger trend magnitudes modelled than identified by Weber et al., which they attribute to particularly low column zoone values at the end of the record consistent with interannual variability. Please see our response to comment 1 for further information.**

Page 15, Figure 2: In the legend in the Figure. Please replace "Model trend" by "simple trend" or "raw trend". That would be clearer, and the "MLR" trends are "model" trends as well. In the caption, please explain how the error bars where obtained.

**Names in the figure legend have been changed to match the figure caption and main text. Extra text has been added to the figure captions for figure 2-4 which provide information on how the error bars were calculated.**

Page 16, Figure 3: In the caption, please explain how the error bars where obtained. See also my major comment 4.

**Extra text has been added to the figure captions for figure 2-4 which provide information on how the error bars were calculated.**

Page 17, Figure 4: Why was that not done for the MLR / residual total ozone as well? Should that not be shown? In the caption, please explain how the error bars where obtained.

**The aim of the final section of the paper is to highlight that at some latitudes the absolute ozone column abundance may not recover to its 1980 values, due in part to the fact that the 1980s were years during a solar maximum, and in part to other factors which overwhelm the recovery trend expected by decreasing CFCs (e.g. increasing BDC speeds resulting from increased GHG concentrations). Further, due to the high interannual variability in the Arctic, it may be possible for years late in the 21$^{st}$ century to have very low column ozone abundancies due to the high natural variability in these regions (see e.g. Bednarz et al., 2016). For these reasons it was felt that discussion of the raw modelled column ozone abundancies was more pertinent than the ozone residuals calculated elsewhere in the manuscript. Extra text has been added to the figure captions for figure 2-4 which provide information on how the error bars were calculated.**

---

## Author Comment (AC2) · 27 Apr 2018

Response to Anonymous Referee #2

General comments:

The expected stratospheric ozone recovery from the effect of halogenated ozone depleting substances (ODSs) has received much attention in recent years. Yet detecting the recovery of the ozone layer is complex due to a number of factors, including internal and external variability, that obscure the emerging signal associated with the slow decline in ODSs levels. The manuscript addresses this issue by investigating three stages of ozone recovery. To this end, the authors use total column ozone (TCO) changes based on experiments of the UM-UKCA and multiple linear regression (MLR) analysis. Although models are not perfect (e.g. often show significant disagreement compared to observations), they are a valuable mean to explore ozone changes due to specific factors (i.e. ODSs levels). Overall, the manuscript addresses relevant issues with regard to the evolution of the stratospheric ozone layer and uses appropriate data and methods. The text is technically well written. I have minor specific comments (detailed below), which I hope will help the authors improve the paper. In general, I suggest more detailed description and evaluation, additional comparison with ozone measurements, and further discussion on existing literature. Therefore, the manuscript is recommended for publication after the specific and technical comments are addressed.

Specific comments:

a. In the Introduction section, the authors clearly set out the stratospheric ozone depletion in the last decades associated with man-made emissions of ODSs. Due to international efforts banning the use of these substances, the ozone layer is expected to recover and the study aims to explore different stages. However, significant work has been done on detection and attribution of ozone recovery, hence it would be appropriate (and helpful for the broader audience) to briefly introduce key findings, remaining issues, and link it with the novelty of this work. Moreover, this will help relate and put into context the main findings here later in the manuscript.

**Reference to the recent findings of Chehade et al. (2012), Pawson et al. (2014) and Weber et al. (2018) has been added to the introduction section in order to further establish the novelty of this work.**

b. In the Model configuration and simulations section (page 3, lines 25–26), the authors explain that the simulations used were performed in support of the CCMI activity, and that are described in more detail in Bednarz et al. (2016) and Keeble et al. (2017). Bednarz et al. (2016) described that the simulations included a future climatological solar cycle since 2009 based on the observed cycle 23,

which is not consistent with the description given in the manuscript (page 3, lines 16–17). Please clarify.

**This is a mistake here – the details of the solar cycle should be those provided by Bednarz et al. (2016) – historic observed solar forcings are applied until 2009, after which cycle 23 is repeated until the end of the simulation. The manuscript has been corrected to account for this. The MLR and results presented in this study are not affected by this error as we use the correct top of atmosphere solar flux prescribed in the model for this analysis.**

c. In the Removing natural cycles section, the text describes a MLR analysis to identify the impacts of natural variability on TCO. Since the results of this study heavily rely on the MLR analysis, I think this section requires more detailed description of the statistical method. In particular, the TO3i and Ne,l,t terms need better description (i.e. "… some constant value" and "Any noise…"). Also, an evaluation of the MLR analysis is important – i.e. How good is it? How much of the model raw data is captured by the MLR and how much "noise" is left? –. The manuscript already includes some references on MLR analysis that may help.

**Further detail has been added to section 3 describing the MLR model and its terms. The TO3i term (now changes to I following advice from reviewer 1) corresponds to the intercept term of the MLR. N corresponds to the month to month variations not accounted for by the other explanatory variables. We feel that the MLR does a good job accounting for the natural cycles we seek to remove, as shown by comparing the red and blue lines in figure 1 and discussed in the manuscript in section 4.**

d. For the Modelled global column ozone and minimum values section, it may be appropriate a statement about the choice of not including the polar regions in this analysis (Figure 1), since, in other sections and figures these regions are included and also discussed in the last paragraph here (page 5, lines 18–25). In fact, the latter paragraph argues that minimum column ozone values are a poor indicator of ozone recovery by giving examples based on polar regions. Is there any particular reason for not using the latest version (3.3) of the Bodeker Scientific database? The latest version, in addition to include some improvements on the methodology, could be expanded until 2016 in the inset of Fig. 1. Also it would be nice to include the Bodeker Scientific database in the acknowledgements, as recommended on the website.

**For figure 1 we present only data from 60S-60N as i) the interannual variability at high latitudes, particularly the Arctic, is very large and identification of longterm changes is more difficult, and ii) high latitude polar ozone depletion is strongly seasonal, and this feature**

**dominates monthly mean time series as the seasonal cycle changes with changing stratospheric ozone depletion. We have clarified in the discussion about minimum column values that we only consider values between 60S-60N.**

**An earlier version of the Bodeker Scientific database (v2.8) was used as this dataset includes monthly mean values, as are presented from our model results. We are reticent to use the latest version as it does not, at present, include monthly mean data, and these would have to be calculated from the daily data provided. This requires a number of decisions which would need to be made (e.g. what spatial and temporal coverage is required for a monthly mean datapoint) which are not trivial and may not match those reached by Bodeker Scientific themselves. In this case, any representation of monthly mean time series we produce may not match the final monthly mean dataset provided by Bodeker Scientific in the future, which may lead to confusion. Bodeker Scientific has been added to the acknowledgements.**

e. Regional trends section. This section includes very interesting results. However, modelled results in Fig. 2, both "raw" and residual data, could be compared to observed trends. In turn, this may lead to some evaluation/discussion and to put into context these results with existing literature. Nevertheless, there is some discussion (outlook) on TCO trends between 2000–2017 in the Discussion and Conclusions section (page 8, lines 15–19). Error bars representing the 95% confidence interval may need a line or two detailing how these are estimated and whether they account for autocorrelation. Are these confidence intervals calculated in the same way for all analyses?

**Additional information on how trend uncertainties are calculated has been added to the manuscript, alongside comparison of our modelled trends with observed trends calculated by Chehade et al. (2012), Pawson et al. (2014) and Weber et al. (2018). Please see our response to review 1 for further information.**

f. Return to historic values section. Figure 4 shows that TCO values in the tropics (<30º) reach the "1980 last recovery" between ~2060s–2070s. However, the main text (page 7, lines 12–14) explains that "…, it is the only region in which total column ozone abundances are not greater than their 1980s values by the end of the simulation,…". Please clarify. Also, is there any particular reason for not showing(using) "ozone residuals" on Fig. 4 as in previous analyses? I understand the study aims to explore ozone recovery addressing natural cycles.

**The aim of the final section of the paper is to highlight that at some latitudes the absolute ozone column abundance may not recover to its 1980 values, due in part to the fact that the 1980s were years during a solar maximum, and in part to other factors which overwhelm the recovery**

**trend expected by decreasing CFCs (e.g. increasing BDC speeds resulting from increased GHG concentrations). Further, due to the high interannual variability in the Arctic, it may be possible for years late in the 21ˢᵗ century to have very low column ozone abundancies due to the high natural variability in these regions (see e.g. Bednarz et al., 2016). For these reasons it was felt that discussion of the raw modelled column ozone abundancies was more pertinent than the ozone residuals calculated elsewhere in the manuscript. Extra text has been added to the figure captions for figure 2-4 which provide information on how the error bars were calculated.**

Technical comments:

Page 1, lines 9–10. "This approach…". The approach or method has not really been introduced. I suggest rephrasing this sentence (e.g. Here internal atmospheric variability… is accounted for by…).

**This sentence has been reworded to read "The impacts of modelled internal atmospheric variability are accounted for by applying a multiple linear regression model to modelled total column ozone values, and ozone trend analysis is performed on the resulting ozone residuals."**

Page 1, line 28. Substitute "ODSs" for "ODS" for consistency throughout the text.

**Substituted**

Page 1, line 34. Randel and Wu (1995) did not explore the effects of Mt Pinatubo eruption on stratospheric ozone.

**This reference should be Randel et al., 1995 (Randel, W. J., Wu, F., Russell, J. M., Waters, J. W., and Froidevaux, L.: Ozone and temperature changes in the stratosphere following the eruption of Mount Pinatubo, J. Geophys. Res., 100, 16753–16764, doi:10.1029/95JD01001, 1995.) and has been corrected in the text and reference list.**

Page 1, lines 37–38. References order.

**References have been reordered.**

Page 1, line 38. Delete "," between "other" and "non-chlorinated".

**Deleted**

Page 2, line 2. References order.

**References have been reordered.**

Page 2, lines 13–15. "Good agreement…" This sentence is a bit confusing, rephrasing maybe?

**This sentence has been rewritten to read:**

**"If good agreement is found between the model and observations when all processes are included, then evidence of ozone recovery due to decreasing stratospheric halogen loadings can be identified by excluding other processes. For example, Solomon et al. (2016) found evidence for healing of the Antarctic ozone layer in September when polar halogen chemistry is included but interannual dynamical variability and volcanic factors are excluded."**

Page 2, lines 22–23. "…, data from fully coupled chemistry-climate model…" is a bit misleading since you use imposed SSTs. I would clarify "fully coupled" (chemistry and radiation schemes?).

**To avoid confusion, the word fully has been removed so that the sentence now reads "To explore future ozone trends and recovery, data from coupled chemistry-climate model (CCM) simulations are required."**

Page 2, line 26. Spell out "SSTs".

**SSTs have been defined here before the first use of SST and later in section 2 just "SSTs" is used.**

Page 2, line 37. Substitute "-" from "–".

**Corrected**

Page3, line 23. Could use just "SSTs", as it was introduced before.

**See comment above**

Page 5, line 33. Substitute "… DU year-1…" for "… TCO (DU year-1)…"? I am aware that "for the column ozone" is mentioned later in the sentence, though it is somehow confusing.

**This sentence has been reworded to read "Figure 2 shows total column ozone trends (in DU year$^{-1}$) obtained from the raw data from the UM-UKCA simulation and the ozone residuals for the decline (1980-1997) and recovery (2000-2017) phases, averaged over 10° latitude bands."**

Page 7, lines 9–10. "… 1980s values for the first time (light red)…" should be "blue".

**We apologise for the confusion – this figure was replotted a number of times. The text now matches the colours in the figure.**

Page 7, line 21. Typo: "airmasses".

**Corrected**

Page 7, line 26. I would substitute "expected" for "projected" (e.g. acknowledging these are modelled results, which are model and scenario dependent).

**This change has been made**

Page 8, line 15. Typo: "… Unlike a recent analyses…"

**This has been corrected to "In contrast to a recent analysis of total ozone measurements…"**

Page 8, line 26. Typo: "… The tropics have too small a trend…"

**This sentence has been altered to read:**
**"The magnitude of the column ozone recovery trend in the tropics is too small in comparison with the natural variability resulting from the solar cycle and the QBO to identify significant trends."**

Page 15, line 2; and Figure 2, legend. Please follow same consistency in the naming, both for the figure and the main text.

**All figures have bee reproduced so that figure legends, figure caption and the main text use consistent naming conventions.**